# Changes in water mass composition and circulation in the central Arctic Ocean between 2011 and 2021 inferred from tracer observations

Anne-Marie Wefing[1,2], Annabel Payne[1], Marcel Scheiwiller[1], Christof Vockenhuber[1,3], Marcus Christl[3], Toste Tanhua[4], and Núria Casacuberta[1,3]

[1]Department of Environmental Systems Science, ETHZ, Zurich, Switzerland
[2]Norwegian Polar Institute, Tromsø, Norway
[3]Laboratory of Ion Beam Physics, ETHZ, Zurich, Switzerland
[4]GEOMAR Helmholtz Centre for Ocean Research Kiel, Kiel, Germany

**Correspondence:** Anne-Marie Wefing (annemarie.wefing@npolar.no)

**Abstract.** The Arctic Ocean is changing rapidly and Atlantic Water circulation plays a key role in the warming, sea-ice decline, and ecosystem changes observed in the Arctic. Still, we only have limited understanding of the pathways and circulation times of Atlantic-derived water both at surface and mid-depth layers in the Arctic Ocean, and their evolution over time. Here, we investigate the water mass composition and circulation in the central Arctic Ocean in 2021 and assess temporal changes thereof

between 2011 and 2021 by using the long-lived anthropogenic radionuclides $^{129}$I and $^{236}$U. This study is based on radionuclide data collected in the central Arctic Ocean and for the first time north of Greenland as part of one of the ocean expeditions of the Synoptic Arctic Survey (SAS-Oden 2021), and available historic data across the Arctic Ocean between 2011 and 2021. We obtain tracer ages as well as the mixing of different endmembers in the surface layer using a tracer-based mixing model. Atlantic Water circulation times and mixing in the mid-depth Atlantic layer are obtained from the Transit Time Distribution

(TTD) model. For 2021, we find a sharp decrease in surface $^{129}$I and $^{236}$U concentrations between the Amundsen and Makarov Basins, pointing to substantial fractions of Pacific Water reaching the Lomonosov Ridge from the Amerasian side. In the halocline layer, similar $^{129}$I and $^{236}$U concentrations on both sides of the Lomonosov Ridge suggest a common formation region of halocline waters with a clear Atlantic Water signal. North of Greenland, we find a mixture of waters that originate from the Canada and Amundsen Basins, both in the surface and the mid-depth layer. Circulation times of Atlantic Water in the mid-

depth layer point to a longer transport route on the Makarov Basin side of the Lomonosov Ridge compared to the Amundsen Basin. When looking at the temporal variability between 2011 and 2021, we observe a shift of the Atlantic-Pacific Water front from the Makarov Basin towards the Lomonosov Ridge from 2011/12 to 2015 and 2021. In the mid-depth Atlantic layer, we find an increase in mean and mode ages from 2015 to 2021, suggesting a slowdown or changes in the pathways of the Arctic Ocean Boundary Current, which is in line with recent studies based on gas tracers.

# 1 Introduction

## 1.1 A Changing Arctic Ocean

The Arctic region is undergoing rapid changes as a result of anthropogenic climate change, including an unprecedented warming trend (Chapman and Walsh, 1993; Serreze et al., 2009; Rantanen et al., 2022), a decrease in sea-ice coverage and thickness (e.g., Kwok, 2018; Sumata et al., 2023), changes in the biogeochemistry including acidification and primary production (e.g., Juranek, 2022), and shifts in the ecosystem (e.g., Ingvaldsen et al., 2021; Kohlbach et al., 2025).

Resulting changes in the freshwater dynamics have emerged as a key focus of current research. Freshwater in the central Arctic Ocean is composed of sea-ice melt, meteoric water, and Pacific Water (the latter being fresher than Atlantic-origin waters). The distribution of freshwater is bound to transport patterns at the surface layer. Changes in the surface layer circulation therefore impact the export of freshwater from the Arctic Ocean to lower latitudes, either through Fram Strait or the Canadian Archipelago (e.g., Azetsu-Scott et al., 2012; Karpouzoglou et al., 2022), or its accumulation within the Arctic Ocean (Proshutinsky et al., 2019). Enhanced freshening of the Arctic and resulting changes in stratification affect sea-ice growth and primary production (Ardyna and Arrigo, 2020). Furthermore, Atlantic and Pacific Waters differ in nutrient concentrations and changes in the spatial distribution hence influence biogeochemistry and ecosystem dynamics (Jones et al., 1998). The observed changes are conveyed beyond the Arctic region, where they ultimately impact the global oceanic circulation. For instance, a freshening of the East Greenland Current, transporting less saline waters southwards through Fram Strait, impacts the surface density in the subpolar North Atlantic and thereby weakens the formation of dense waters and the overturning circulation at lower latitudes (e.g., Aagaard et al., 1985; Manabe and Stouffer, 1995; Rahmstorf, 1996; Sévellec et al., 2017; Brakstad et al., 2019; Le Bras et al., 2021; Abot et al., 2023).

Part of the observed changes in the Arctic Ocean are linked to the expansion of warm Atlantic inflows into the Eurasian Basin ("Atlantification"), associated with weakening stratification, a thinned halocline layer, and enhanced vertical mixing (Polyakov et al., 2005, 2017, 2020; Wang et al., 2024). The recent study by Polyakov et al. (2025) showed that this Atlantification is not limited to the Eurasian Arctic anymore, but also observed beyond the Lomonosov Ridge, in the Makarov Basin. With respect to biogeochemistry, the shift to more Atlantic-like conditions might increase nutrient availability and potentially also support projected increases in primary production (Henley et al., 2020). In the mid-depth Atlantic Water, changes in circulation and associated timescales affect the storage and transport of anthropogenic Carbon in the Arctic (e.g., Raimondi et al., 2024).

Temporal changes in both surface and mid-depth layer circulation are associated with shifts in atmospheric circulation patterns and coupled oceanic responses, described by indices such as the Arctic Oscillation (AO; e.g., Morison et al., 2012, 2021), the Arctic Ocean Oscillation (AOO; e.g., Proshutinsky et al., 2015), or the Arctic Dipole (AD; e.g., Polyakov et al., 2023). An increased AO index is for instance linked to an increase in the cyclonic atmospheric circulation of the Northern Hemisphere and a low Arctic sea level atmospheric pressure. It was therefore also termed the "cyclonic mode" and found to induce shifts

in freshwater pathways and in the front between Atlantic- and Pacific-derived waters in the surface layer of the central Arctic Ocean (Morison et al., 2012; Smith et al., 2021). A better understanding of the water mass composition and circulation patterns in the Arctic Ocean, as well as temporal changes thereof, is crucial to monitor ongoing changes and improve future projections.

## 1.2   Arctic Ocean Circulation and Vertical Structure

In the cold and fresh surface layer of the Arctic Ocean, here referred to as Polar Surface Water (PSW), the two primary large-scale circulation patterns are the Beaufort Gyre and the Transpolar Drift. The Beaufort Gyre is an anticyclonic circulation system in the Canada Basin (e.g., Aagaard and Carmack, 1989; Carmack, 2000; Timmermans and Toole, 2023). The Transpolar Drift transports surface water and sea ice from the East Siberian Shelf and the Laptev Sea to the Fram Strait (e.g., Morison et al., 2012). Below the surface layer, warm and saline Atlantic-origin water forms the mid-depth Atlantic layer of the Arctic Ocean. Atlantic Water enters via two gateways, Fram Strait and the Barents Sea. Both branches are cooled along their northward passage and converge in the St. Anna Trough region. From here, the Fram Strait Branch Water (FSBW) and the Barents Sea Branch Water (BSBW) circulate cyclonically through the Arctic Ocean along different loops, forming the Arctic Ocean Boundary Current (AOBC). Part of the Atlantic Water recirculates to Fram Strait within the Eurasian Basin and part of it enters the Amerasian Basin, following a longer loop before also exiting through Fram Strait (Fig. 1; e.g., Anderson and Jones, 1992; Schauer et al., 2002).

The PSW layer is highly stratified and generally entails Pacific Water, river runoff, sea-ice meltwater, as well as transformed Atlantic Water. Pacific Water enters the Arctic Ocean through Bering Strait, mainly resides in the Canada and Makarov Basins, and is restricted to the PSW layer due to its low density (e.g., Coachman and Barnes, 1961; Rudels, 2015) (Fig. 1). PSW can be further divided into the Polar Mixed Layer (PML) at the surface, reaching down to the temperature minimum, which is a remnant of the previous winter convection (Rudels et al., 1996), and the halocline layer. In the Eurasian Basin, the halocline layer consists of the lower halocline (LHC), which originates from the cooling and freshening of Atlantic Water north of Fram Strait and the Barents Sea (Rudels et al., 1996; Korhonen et al., 2013). In the Amerasian Basin, the halocline layer consists of the LHC and an additional layer above, the upper halocline (UHC). The UHC mainly consists of low salinity Pacific Water and Siberian river runoff (Jones and Anderson, 1986; Anderson et al., 1994; Rudels et al., 2004). Below the PSW, Atlantic Water with temperatures above $0°C$ resides on top of intermediate and deep waters.

## 1.3   Transient Tracers to study Atlantic Water Circulation in the Arctic Ocean

The role of Atlantic Water circulation in the Arctic Ocean and associated changes have been investigated through various methods, including modeling studies (e.g., Wang et al., 2024), hydrographic measurements (e.g., Woodgate et al., 2001; Polyakov et al., 2005; Dmitrenko et al., 2008; Li et al., 2021; Schulz et al., 2024), as well as tracer studies (e.g., Tanhua et al., 2009; Casacuberta and Smith, 2023; Gerke et al., 2024; Körtke et al., 2024). Classically, gas tracers such as CFCs and $SF_6$ have

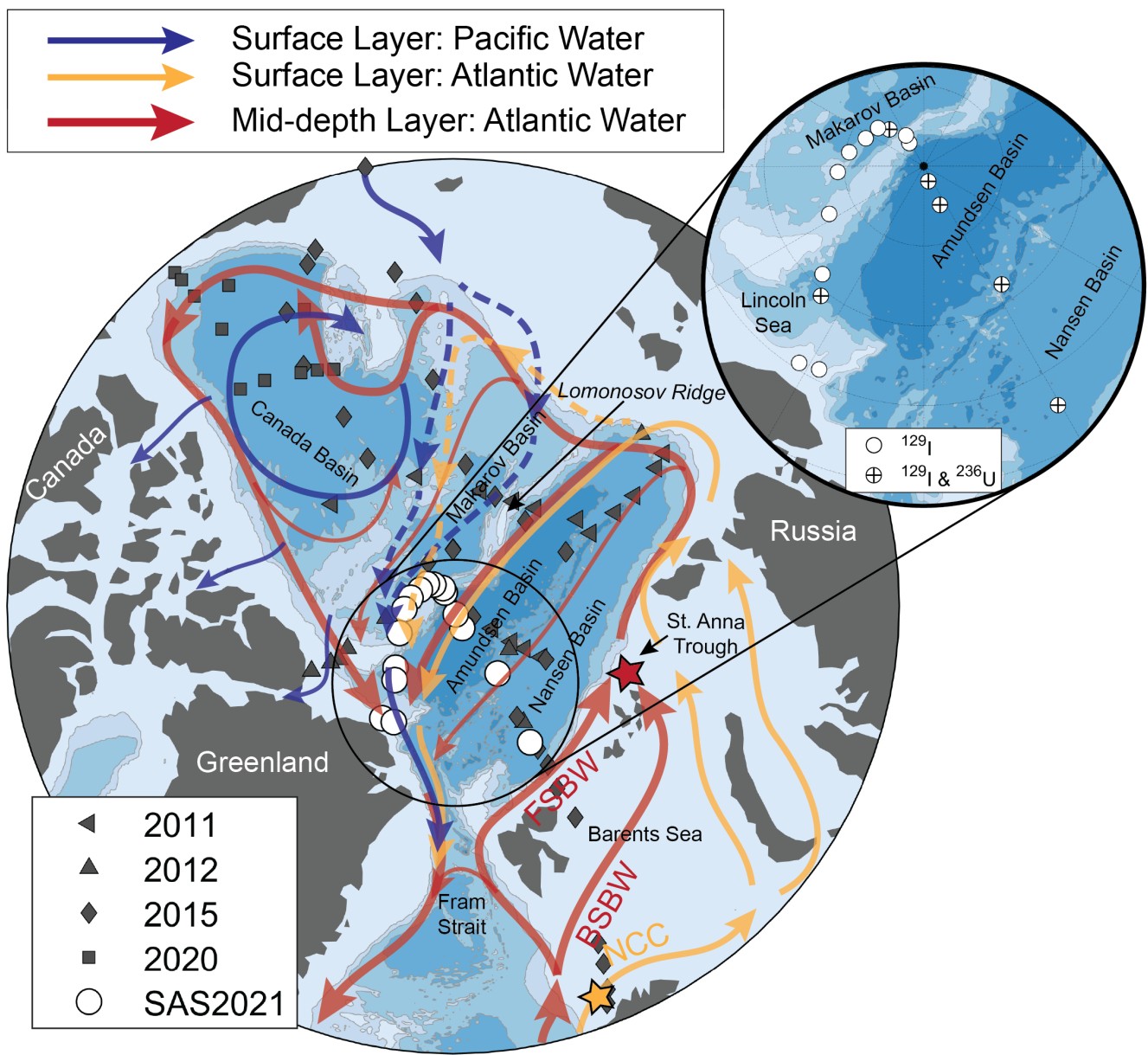

**Figure 1.** Map of the Arctic Ocean with main topographic and circulation features, as well as stations of radionuclide samples considered in this study. Blue and orange arrows show circulation patterns of Pacific (e.g., McLaughlin et al., 1996) and Atlantic Water (e.g., Rudels et al., 2004) in the surface, respectively, red arrows show circulation of Atlantic Water at mid-depth. FSBW: Fram Strait Branch Water, BSBW: Barents Sea Branch Water, NCC: Norwegian Coastal Current. The yellow and red star denote the locations where the surface layer input function (yellow, southern Barents Sea Opening) and mid-depth Atlantic layer input function (red, St. Anna Trough) were defined (see Fig. 2). Stations with published $^{129}$I and $^{236}$U data from earlier years (2011-2020) are shown in dark grey with different symbols for different sampling years, see Section 2.1 for details. Radionuclide stations from SAS2021 are shown in white, with the inset further distinguishing stations where only $^{129}$I data is available (white circles) and stations where $^{129}$I and $^{236}$U data is available (white circles with black cross).

been used to this purpose (e.g., Frank et al., 1998; Smethie et al., 2000). Since these tracers are introduced globally by air-sea gas exchange, studies focused on ventilation timescales. The recent study by Gerke et al. (2024) based on CFC-12 and SF$_6$ data from the Eurasian Basin collected between 1991 and 2021 found higher mean ages in the Amundsen Basin in 2021 compared to 2005 - 2015. This implies a decrease in ventilation, which was found to have occurred primarily between 2005 and 2021. The authors speculate about a possible decrease in the strength of the boundary current, increasing transport times of Atlantic Water. Körtke et al. (2024) investigated temporal changes in Atlantic Water pathways in the Arctic Ocean based on tracer ages obtained from CFC-12 and SF$_6$. Lower Atlantic Water tracer ages in the mid-1990s were attributed to a strong boundary current connected to a positive AO index (see also Morison et al., 2021). Elevated tracer ages in 2005 and 2015 were interpreted as a weakening of the boundary current and coincided with phases of largely negative or mixed AO index. In addition, tracer age changes were also attributed to changes in the Atlantic Water pathways (see, e.g., Fig. 7 in Körtke et al., 2024).

Apart from gas tracers, anthropogenic radionuclides labeling Atlantic Water have proven to be valuable tools (e.g., Smith et al., 2011, 2021; Casacuberta and Smith, 2023). In recent years, the two radionuclides $^{129}$I and $^{236}$U, mainly introduced into the Arctic Ocean as liquid releases from European reprocessing plants, have been utilized as Atlantic Water tracers in several studies. The combination of $^{129}$I and $^{236}$U provided a better understanding of Atlantic Water pathways (Casacuberta et al., 2018), Atlantic Water circulation times and changes therein (Wefing et al., 2021), mixing within the flow field (Payne et al., 2024), the uptake of anthropogenic carbon by Atlantic Waters (Raimondi et al., 2024), as well as mixing of different water masses and freshwater components in waters outflowing the Arctic through Fram Strait (Wefing et al., 2022).

$^{129}$I and $^{236}$U are most powerful in the Arctic Ocean and the subpolar North Atlantic, downstream from the point-like input from the reprocessing plants. In contrast to CFCs and SF$_6$, $^{129}$I and $^{236}$U can also be used to assess circulation times in the surface layer, which is in contact with the atmosphere, as they are introduced mainly via liquid discharges and not through air-sea gas exchange (Smith et al., 2011; Wefing et al., 2021). For the comparison of gas tracers and radionuclides, the Arctic Ocean provides a unique configuration, since Atlantic Waters are isolated from atmospheric exchange once they subduct to depths and are capped by the halocline layer (see also Raimondi et al. (2024) for further discussion). With the rapid changes occurring in the Arctic Ocean, on the order of decades or below (e.g., Solomon et al., 2021; Polyakov et al., 2025), available historical $^{129}$I and $^{236}$U data can now be used to assess temporal changes in Atlantic Water circulation (Smith et al., 2021), complementing tracer-based, hydrographic, and modeling studies.

This study aims to assess the circulation pathways and timescales of Atlantic Water in the central Arctic Ocean in 2021, with particular focus on the Lincoln Sea north of Greenland - a strategic location where waters from the Eurasian and Amerasian Basin converge before exiting the Arctic through the Nares or Fram Strait (Newton and Sotirin, 1997; de Steur et al., 2013). By combining new $^{129}$I and $^{236}$U data collected in 2021 with historical data from similar locations, we constrain transport times and mixing processes of Atlantic-derived waters, characterize the composition of surface waters with particular emphasis on the extent of Pacific Water, and evaluate temporal changes in circulation over the decade from 2011 to 2021. This study contributes

to the understanding of how changes in the Atlantic Waters entering the Arctic Ocean affect the circulation in the Arctic and how these waters mix with Pacific-origin waters in the upper water column.


## 2 Material and Methods

### 2.1 Radionuclide Tracer Data

Seawater samples for the analysis of $^{129}$I and $^{236}$U were collected in 2021 during the "SAS-Oden 2021" expedition (SAS2021) aboard the Swedish research icebreaker *Oden* (Snoeijs-Leijonmalm et al., 2022), which was part of the Synoptic Arctic Survey
(SAS). Hydrographic data from CTD profiles for this expedition is available on PANGAEA (Heuzé et al., 2022a). Biogeochemical bottle data is available both on PANGAEA (Heuzé et al., 2022b) and published in GLODAPv2.2023 (Lauvset et al., 2024). In total, 167 samples (16 full-depth stations) were taken for $^{129}$I analysis and 48 samples (6 full-depth stations) were taken for $^{236}$U analysis at several stations along the cruise track from north of Svalbard towards the North Pole and further towards northern Greenland (Fig. 1). For this study, we divided the stations sampled for $^{129}$I and $^{236}$U into two sections: Section 1
comprises stations 5-26 from the Nansen and Amundsen Basins up to the Lomonosov Ridge, Section 2 comprises stations 28-52 from the Makarov Basin and the region north of Greenland. During the expedition, samples were filled directly from the Niskin bottles mounted on the CTD-rosette into rinsed plastic 250 mL bottles (for $^{129}$I analysis) and 3 L plastic cubitainers (for $^{236}$U analysis) and shipped to ETH Zurich, Switzerland, for radionuclide analysis.

$^{129}$I analysis followed the method of Casacuberta et al. (2016). Samples were measured by Accelerator Mass Spectrometry (AMS) using the 0.5 MV AMS system *TANDY* at the Laboratory of Ion Beam Physics at ETH Zurich, Switzerland (Vockenhuber et al., 2015). Concentrated and diluted forms of an in-house standard, C2, were included with each measurement run (nominal value diluted: $^{129}$I/$^{127}$I $= 5.055 \times 10^{-12}$ at at$^{-1}$, nominal value concentrated: $^{129}$I/$^{127}$I $= 38.995 \times 10^{-12}$ at at$^{-1}$). All samples were corrected with chemistry blanks ($n = 12$) prepared in the lab at ETH Zurich (MilliQ water, 18.2 M$\Omega$). Relative
uncertainties are largely around 3 %, based on the analytical uncertainty of the AMS measurement and repeated measurements of an ETH internal standard (standard deviation of 2 %, $n = 6$).

$^{236}$U analysis followed the method of Christl et al. (2015a). Samples were measured using the 0.3 MV AMS system *MILEA* at the Laboratory of Ion Beam Physics at ETH Zurich, Switzerland (Christl et al., 2023). Measured isotope ratios
were normalized to the in-house standard ZUTRI (nominal values: $233$U/$^{238}$U $= (33,170 \pm 830) \times 10^{-12}$ at at$^{-1}$, $^{236}$U/$^{238}$U $= (4,055 \pm 200) \times 10^{-12}$ at at$^{-1}$) (Christl et al., 2013). $^{236}$U and $^{238}$U concentrations were calculated from the measured concentration of the U233 spike. All samples were corrected with chemistry blanks ($n = 5$) prepared in the lab at ETH Zurich (MilliQ water, 18.2 M$\Omega$). Relative uncertainties are largely around 3 %, based on the analytical uncertainty of the AMS measurement and assuming the same standard deviation of repeated measurements as for $^{129}$I.


**Table 1.** Information on published radionuclide data from earlier expeditions used in this study.

| Year | Cruise | Ship | Tracers | References |
|------|--------|------|---------|------------|
| 2011 | PS78 | RV Polarstern | $^{129}$I, $^{236}$U | Casacuberta et al. (2016); Wefing et al. (2021) |
| 2012 | PS80 | RV Polarstern | $^{129}$I, $^{236}$U | Casacuberta et al. (2016); Wefing et al. (2021) |
| 2012 | Switchyard/LDEO | *aircraft* | $^{129}$I | Smith et al. (2021); Casacuberta and Smith (2023) |
| 2015 | PS94 | RV Polarstern | $^{129}$I, $^{236}$U | Casacuberta et al. (2018); Wefing et al. (2021); Raimondi et al. (2024) |
| 2015 | HLY1502 | USCGC Healy | $^{129}$I, $^{236}$U | Smith et al. (2021); Chamizo et al. (2022); Raimondi et al. (2024) |
| 2020 | JOIS2020 | CCGS Louis St Laurent | $^{129}$I, $^{236}$U | Payne et al. (2024) |

$^{129}$I and $^{236}$U data from SAS2021 are available on Zenodo (Wefing, 2025). To assess temporal changes over the past decade, we used published radionuclide data from six past expeditions to the Arctic Ocean (Table 1).

## 2.2 Sources and Input Functions of $^{129}$I and $^{236}$U to the Arctic Ocean

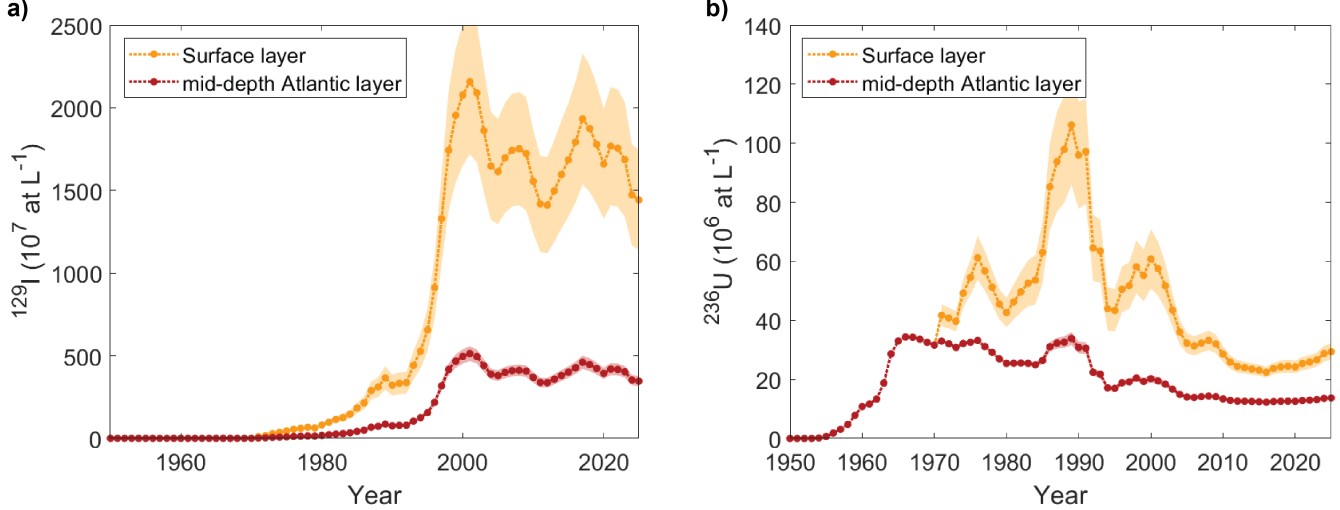

**Figure 2.** Input functions of $^{129}$I (a) and $^{236}$U (b) for the surface and mid-depth Atlantic layer, defined for the northern Norwegian Coast and the St. Anna Trough, respectively. Locations are marked with stars in Fig. 1. Shaded areas are the uncertainties of each input function.

$^{129}$I and $^{236}$U are long-lived radionuclides with half-lives of about 16 Myr and 23 Myr, respectively, and negligible natural background concentrations in the marine environment (Smith et al., 1998, 1999, 2005; Steier et al., 2008; Sakaguchi et al., 2012; Casacuberta et al., 2016). They have been introduced into the ocean by two anthropogenic sources. Firstly, liquid

releases from nuclear reprocessing plants, which introduced mainly $^{129}$I, and secondly, via global fallout from atmospheric nuclear weapon tests, which introduced mainly $^{236}$U (Casacuberta et al., 2016). Although the nuclear weapon tests introduced

significant amounts of $^{236}$U (Sakaguchi et al., 2009), its global dispersion resulted in a low and relatively uniform background level that is now considered constant. In the Arctic Ocean, the dominant sources of $^{129}$I and $^{236}$U are therefore the liquid releases from the two European reprocessing plants, Sellafield in the UK, and La Hague in France.

The global fallout background signal ($7 \times 10^7$ at L$^{-1}$ for $^{129}$I, Cooper et al. (2001), and $6.25 \times 10^6$ at L$^{-1}$ for $^{236}$U, Chamizo
et al. (2022)) is found in all water masses and introduced into the Arctic Ocean by both Atlantic and Pacific Water. The reprocessing plant releases are transported northwards from the North Sea to the Arctic Ocean mainly by the Norwegian Coastal Current (NCC) (Edmonds, 2001; Gascard et al., 2004; Smith et al., 2011; Christl et al., 2015b, Fig 1). The tracer signal carried by the NCC is mixed into surrounding Atlantic Water during its northward passage, and hence all Atlantic Water entering the Arctic Ocean is tagged with a distinct tracer signature of $^{129}$I and $^{236}$U (Casacuberta et al., 2018).


Input functions of $^{129}$I and $^{236}$U describe the concentrations of both tracers entering the Arctic Ocean at a defined initialization point over time (Fig. 2). Here, we used the input functions defined and described in Wefing et al. (2021). Briefly, one input function for each radionuclide was defined for the surface layer, describing the concentrations of both tracers in Atlantic-origin Water that is transported with the NCC over the shelf seas (Barents, Kara, and Laptev Sea) and enters into the Polar Surface
Water. The initialization point for the surface layer input functions lies at around 72°N at the Barents Sea Opening, north of the Norwegian Coast (orange star in Fig. 1). A different input function for $^{129}$I and $^{236}$U was defined for the Atlantic layer, describing the concentrations of both tracers in Atlantic Water (FSBW and BSBW), forming the mid-depth Atlantic-origin layer of the Arctic Ocean. For the Atlantic layer input function, we consider the St. Anna Trough as the initialization point, since this is the location where FSBW and BSBW subduct and jointly form the Atlantic layer further downstream in the Arctic
Ocean (red star in Fig. 1).

### 2.3 Water Mass Classification

Water mass classifications used in this study followed the definition by Marnela et al. (2008) for the general water masses and (Korhonen et al., 2013) for the PSW (PML, UHC, LHC). Samples were generally divided into Polar Surface Water (PSW, $\sigma_\Theta < 27.70$) and Arctic Atlantic Water (AAW, $27.70 \le \sigma_\Theta < 27.97$), as well as the intermediate ($27.97 \le \sigma_\Theta$, $\sigma_{0.5} < 30.444$),
and deep ($30.444 \le \sigma_{0.5}$) layer. The PSW samples were further divided into the Polar Mixed Layer (PML, lower boundary defined by the temperature minimum) and the halocline layer. For samples from the Eurasian Basin, we only considered the lower halocline layer (LHC, from the lower boundary of the PML down to $\Theta = 0°$ C). For samples from the Amerasian Basin, we considered an additional upper halocline layer between PML and LHC as described in Section 1.1 (UHC, from the lower boundary of the PML down to $S_p = 34$).


## 2.4 Tracer Ages and Dilution from Binary Mixing Model

For samples from PSW, a simple binary mixing model with $^{129}$I and $^{236}$U, assuming purely advective flow, was applied to determine radionuclide-based tracer ages and to investigate mixing between different end-members. This model calculates mixing between the surface layer tracer input function and the steady-state global fallout signal in $^{236}$U-$^{129}$I tracer space. Each binary mixing line reflects the dilution of the tracer signal carried into the Arctic Ocean by Atlantic Water in a specific year (described by the input function), with waters carrying the global fallout signal. The latter can either reflect a fraction of Atlantic Water not tagged with the reprocessing plants' tracer signal (Section 2.2) or Pacific Water that entered the Arctic Ocean through Bering Strait. Tracer ages of Atlantic Water (i.e., the time it took a water parcel to travel from the initialization point of the input function to the sampling location) were determined by finding the binary mixing line closest to the sample and correspond to the difference between sampling year and the input function year of that mixing line. Earlier studies have shown that this simple model yields reasonable results for circulation times in the surface layer of the Arctic Ocean (Smith et al., 2011; Casacuberta et al., 2018; Wefing et al., 2021).

For samples from the mid-depth Atlantic Layer and deeper, the binary mixing model was used solely to infer information on mixing and water mass provenance, as circulation times were obtained from the Transit Time Distribution (TTD) method.

## 2.5 Transit Time Distribution Model

The Transit Time Distribution (TTD) method was applied to describe the characteristics of Atlantic Water circulation in the mid-depth layer (samples from 300-1000 m depth) of the Arctic Ocean. This model has mainly been applied to ocean interior ventilation studies (e.g., Haine and Hall, 2002; Waugh et al., 2003; Tanhua et al., 2009; Stöven et al., 2015) and to determine anthropogenic $CO_2$ uptake by the ocean (e.g., Waugh et al., 2006; Khatiwala et al., 2013; Stöven and Tanhua, 2014; Stöven et al., 2016; Raimondi et al., 2024; Gerke et al., 2024). Its use with anthropogenic radionuclides has been investigated in several recent studies (Smith et al., 2011; Wefing et al., 2021; Smith et al., 2021) and the application to the $^{129}$I-$^{236}$U tracer pair has been extensively described in Raimondi et al. (2024, therein referred to as "Smith's TTD approach"). Therefore, we will only provide a brief summary here.

In the TTD model, the concentration of a stable tracer at sampling location $x$ and time $t$ is described by

$$c(x,t) = \int\limits_{0}^{\infty} c_0\left(t - t'\right) G\left(x, t'\right) dt' \tag{1}$$

where $c_0$ represents the tracer input function and $G(x,t)$ is the Green's function, which describes the propagation of the tracer signal and, therefore, the characteristics of the flow from the initialization of the input function to the sampling location. $G$ is essentially a probability density function (PDF) reflecting the weight of tracer signals from different years of the input function at the specified sampling location and time. The flow is limited to 1D for simplicity, and in this case, the PDF is given by an

inverse Gaussian function:

$$G(x,t) = \sqrt{\frac{\Gamma^3}{4\pi\Delta^2 t^3}} \, exp\left(-\frac{\Gamma(t-\Gamma)^2}{4\Delta^2 t}\right) \tag{2}$$

defined by two parameters, $\Gamma$ and $\Delta$. $\Gamma$ represents the mean age of the distribution while $\Delta$ is a measure of the width of the PDF, and hence describes how much a tracer signal disperses during the flow as a result of lateral mixing. $\Delta = 0$ corresponds to purely advective flow. Another age measure of the distribution is the mode age $t_{mode}$ (also termed most probable age, Smith et al., 2011; Wefing et al., 2021), the circulation time with the highest probability within the PDF. It is given by

$$t_{mode} = \frac{1}{\Gamma}\left(\sqrt{9\Delta^4 + \Gamma^4} - 3\Delta^2\right) \text{ with } t_{mode} \geq 1. \tag{3}$$

Previous studies suggested the mode age as a more suitable age measure for the lateral transport of Atlantic Water in the Arctic Ocean compared to the mean age (Smith et al., 2011; Wefing et al., 2021).

In the case of a mixture of two water masses with different transport histories that can each be described by a TTD, we considered a linear combination of two inverse Gaussian TTDs $G_1\left(\Gamma_1, \Delta_1\right)$ and $G_2\left(\Gamma_2, \Delta_2\right)$, each defining the flow field in

the respective branch (Waugh et al., 2003; Smith et al., 2022):

$$G_{mix} = \alpha G_1\left(\Gamma_1, \Delta_1\right) + (1-\alpha)G_2\left(\Gamma_2, \Delta_2\right) \tag{4}$$

where $\alpha$ describes the fraction of the water mass described by $G_1\left(\Gamma_1, \Delta_1\right)$ in the mixture. The mean age of this bimodal TTD is a linear combination of the mean ages of $G_1$ and $G_2$. $\Delta$s do not mix conservatively, but are always higher in the mixture

compared to a linear combination of the two end-members (see, e.g., Fig. 2b in Smith et al., 2022).

In this study, the TTD method was applied to samples not classified as PSW and down to 1000 m. Deeper samples (1500 m and below) had $^{236}$U concentrations close to the global fallout level or below (Fig. 3), and hence they did not carry significant amounts of the transient reprocessing plants tracer signal. The $\Delta - \Gamma$ grid for the determination of both TTD parameters ac-

cording to "Smith's TTD approach" is shown in the appendix (Fig. A1).

## 3   Results

In the results section, we will present the distribution of $^{129}$I and $^{236}$U from SAS2021 in depth profiles (Fig. 3), along the SAS2021 transect as section plots of the upper 500 m (Fig. 4) and in relation to hydrographic parameters (Fig. 5). We restrict

the latter to $^{129}$I, due to the larger dataset compared to $^{236}$U. We will then show tracer ages and dilution obtained from the binary mixing model for the PSW samples. Lastly, we will depict transport times and mixing from the TTD model for samples from the mid-depth Atlantic layer, as well as temporal changes in this layer.

## 3.1 Distribution of $^{129}$I and $^{236}$U

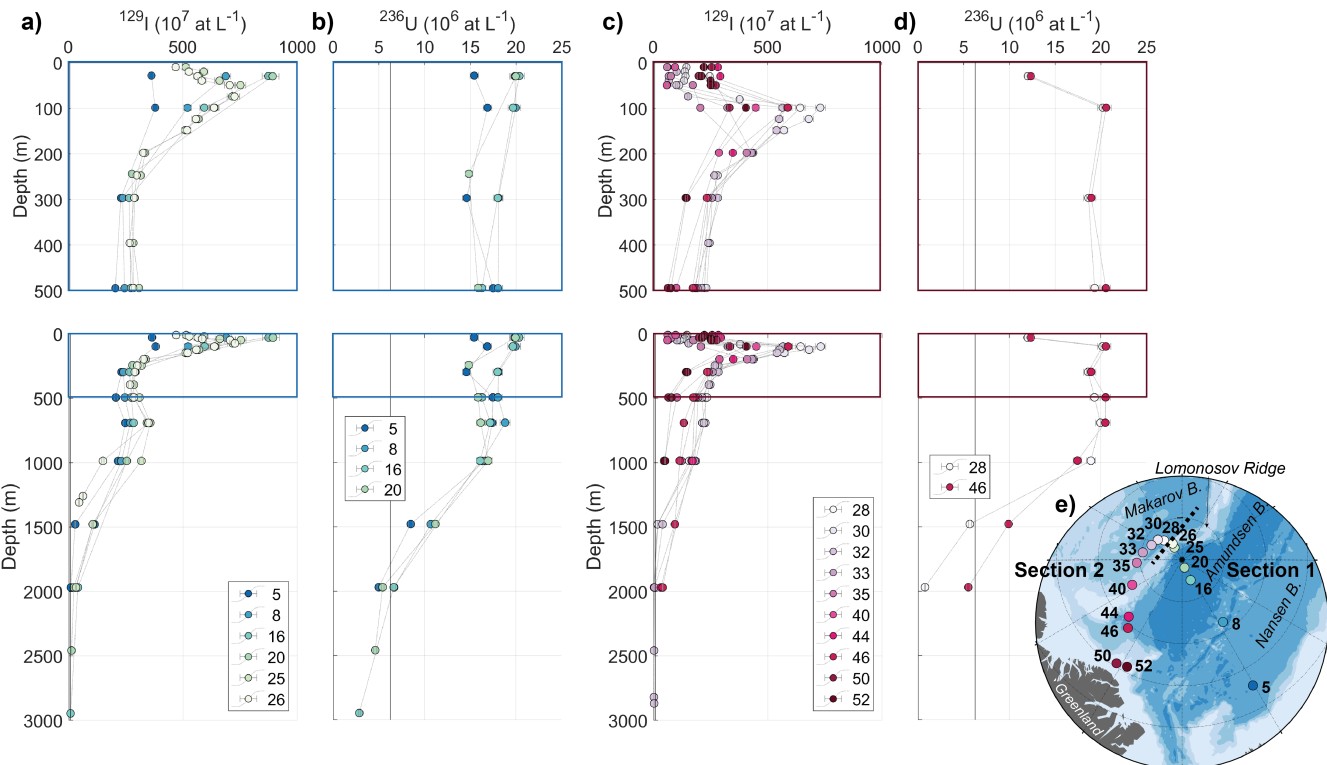

**Figure 3.** Depth profiles of $^{236}$U and $^{129}$I concentrations for all SAS2021 stations, divided into Section 1 (a: $^{129}$I, b: $^{236}$U) and Section 2 (c: $^{129}$I, d: $^{236}$U). Top row is a close up of 0-500 m, marked by the box in the full-depth profiles in the bottom row. Black vertical lines indicate global fallout background signal for both radionuclides from Payne et al. (2024), which is not distinguishable from zero in the $^{129}$I profiles (panels a and c). (e) Map showing stations classified as Section 1 (blue-green colors) and 2 (red-purple colors). The dashed line indicates the division between both sections.

For all profiles from the Section 1 (Eurasian Basin), the highest $^{129}$I concentrations were observed in the upper 100 m, with a peak in $^{129}$I at 50-100 m for stations 25 and 26 (around $700 \times 10^7$ at $L^{-1}$, higher depth resolution for both stations compared to the other stations in section 1) and a peak at around 50 m for stations 16 and 20 (around $900 \times 10^7$ at $L^{-1}$) (Fig. 3a). At all stations, $^{129}$I concentrations decreased with depth down to 500 m and increased again to a local maximum around 700 m (between $200 \times 10^7$ at $L^{-1}$ and $400 \times 10^7$ at $L^{-1}$). Below 700 m, concentrations decreased further to almost 0 in the deepest samples at 3000 m depth. Generally, within the upper 1000 m, higher concentrations were observed at stations further north. Station 5 (located in the Nansen Basin) showed the lowest $^{129}$I concentrations throughout the entire depth range (except for station 26 at 1000 m and below), especially in the upper 100 m. $^{129}$I concentrations at Station 8 (at the Gakkel Ridge) were between those from station 5 and station 16. Station 26 (on top of the Lomonosov Ridge) showed the highest $^{129}$I concentrations

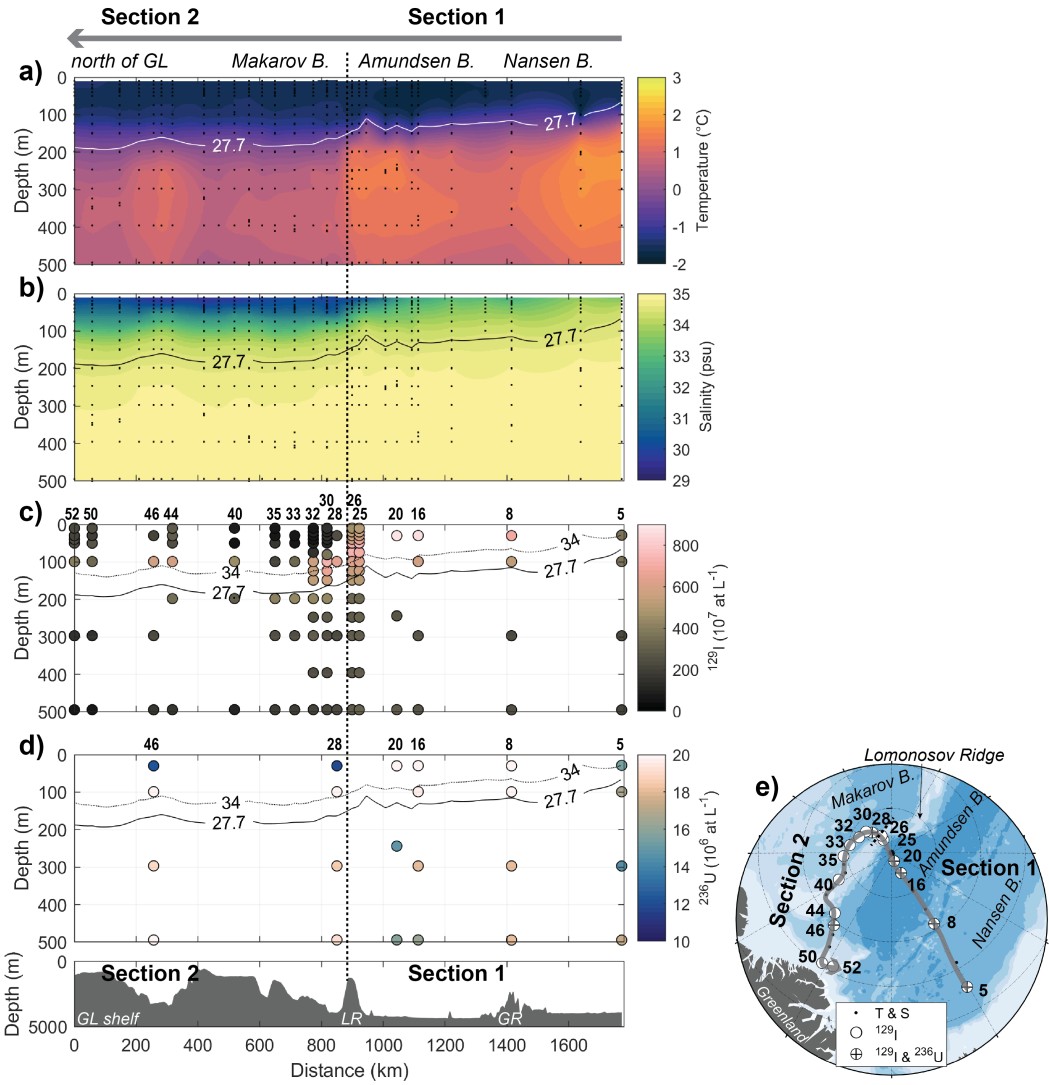

**Figure 4.** Section plots (upper 500 m) of CTD temperature (a), practical salinity (b), $^{129}I$ (c), and $^{236}U$ concentration (d), following the cruise track from the Nansen Basin (right-hand side of plots) to the Lomonosov Ridge (middle) and the Greenland shelf (left-hand side of plots). Continuous line is $\sigma_\Theta = 27.7$, dashed line in c and d is the isoline of practical salinity 34. Seafloor topography from GEBCO (GEBCO Compilation Group, 2024) is shown below the section plots, indicating the Greenland shelf (GL shelf), the Lomonosov Ridge (LR), and the Gakkel Ridge (GR). The transect is indicated as a grey line in the station map (e).

among all Section 1 stations between depths of 100 to 700 m, but concentrations decreased sharply between 700 and 1000 m. In contrast to $^{129}$I, $^{236}$U concentrations across Section 1 (sampled in lower spatial and depth resolution compared to $^{129}$I) did not exhibit much variability in the upper 1000 m ($15 - 20 \times 10^6$ at L$^{-1}$) and no clear trend was observed among different stations (Fig. 3b). No apparent peak in $^{236}$U concentrations was observed at 50-100 m, however, the sampling resolution was lower compared to $^{129}$I. Similar to $^{129}$I, a slight local maximum was observed at 700 m depth. $^{236}$U concentrations reached global fallout levels at around 2000 m depth and decreased further to around $3 \times 10^6$ at L$^{-1}$ at 3000 m depth.

In Section 2 (Makarov Basin and north of Greenland), the maximum of $^{129}$I concentrations ($200 - 700 \times 10^7$ at L$^{-1}$) was found around 100 m depth for all Section 2 stations, therefore, at greater depths than for Section 1 (Fig. 3c). Towards the surface, concentrations decreased to around $100 - 300 \times 10^7$ at L$^{-1}$. Surface $^{129}$I concentrations of all Section 2 stations were lower than those from Section 1, and the overall lowest $^{129}$I concentrations were observed for stations in the Makarov basin. Generally, as for Section 1, profiles further to the north exhibited higher concentrations (above 1000 m). The two available $^{236}$U profiles from Section 2 were very similar in the upper 1000 m (Fig. 3d). Above 100 m depth, $^{236}$U concentrations decreased towards the surface. In contrast, they were almost constant over the depth range of 100-1000 m. Station 28 (in the Makarov Basin) showed a substantially lower $^{236}$U concentration (almost 0) at 2000 m than all other stations (including those from Section 1).

At the Lomonosov Ridge (brightest colors in Fig. 3), we observed a clear drop in $^{129}$I from station 26 to 28, i.e., from the Amundsen Basin to the Makarov Basin, at depths of around 50 m (decrease of about 50 %) and between 500 and 1000 m (especially at 700 m, decrease of about 35 %). In contrast, between 50 and 500 m depth, concentrations were similar at both stations. Differences in upper layer $^{129}$I and $^{236}$U between Section 1 and Section 2 were also visible in the section plots covering both sections (Fig. 4). Above 100 m, the decline in $^{129}$I and $^{236}$U in Section 2 was evident and coincided with a decrease in practical salinity from around 32.5 to around 30.5 at 50 m depth.

## 3.2 $^{129}$I in Relation to Hydrographic Parameters

In the T-S plot (Fig. 5a, potential temperature against practical salinity, different water masses are highlighted following Section 2.3) the two sections of the expedition showed different features: PSW from Section 2 had higher potential temperatures (difference around $0.5°$C) compared to Section 1, for practical salinities above 32 (see also Section 4.2). In AAW and below, the opposite was observed, with Section 1 stations showing higher potential temperatures (difference up to $1°$C) compared to Section 2 (Fig. 5b). Highest potential temperatures in AAW of about $1.5°$C were observed at station 5 in the Nansen Basin, closest to the Atlantic Water inflow region. $^{129}$I was high throughout the entire PSW in Section 1 and highest in the LHC for Section 2.

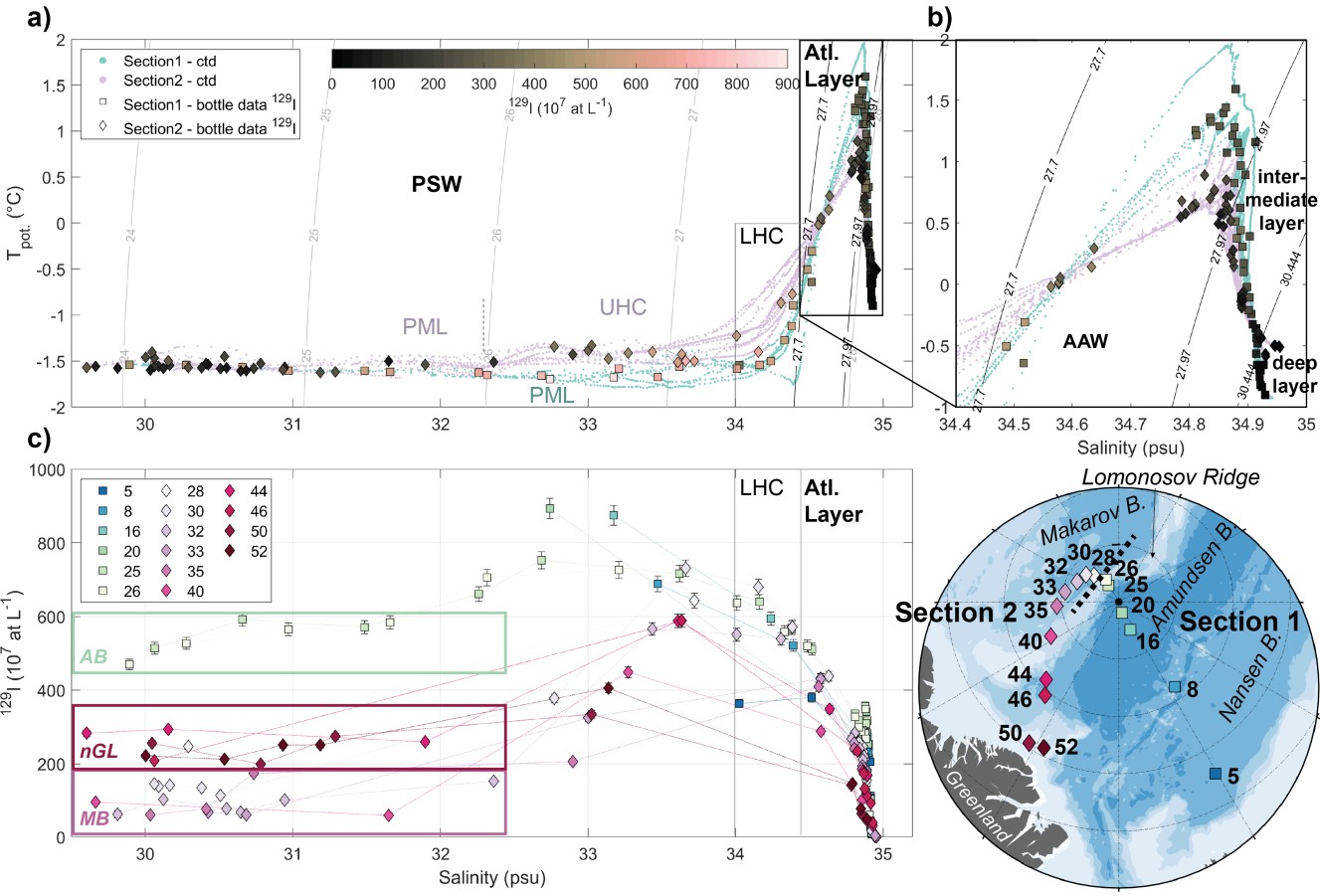

**Figure 5.** (a) T-S plot (potential temperature against practical salinity) with $^{129}$I concentrations color-coded and different water masses indicated (PSW: Polar Surface Water, PML: Polar Mixed Layer, UHC: upper halocline, LHC: lower halocline, AAW: Arctic Atlantic Water). Teal and purple dots are 1 m-binned CTD data from Section 1 and 2, respectively. Square and diamond data points are bottle file data from Section 1 and 2, respectively. (b) close-up of the Atlantic Layer (black box in a). (c) Plot of $^{129}$I concentration against practical salinity with stations color-coded. AB: Amundsen Basin, nGL: north of Greenland, MB: Makarov Basin.

Different sampling regions showed different relations between $^{129}$I and practical salinity (Fig. 5c). At practical salinities $< 32$ (PML), highest $^{129}$I concentrations ($400 - 600 \times 10^7$ at L$^{-1}$) were found in samples from the Amundsen Basin (Section 1), followed by the stations north of Greenland (Section 2, $200 - 300 \times 10^7$ at L$^{-1}$). Makarov Basin stations (also Section 2) exhibited the lowest $^{129}$I concentrations ($0 - 200 \times 10^7$ at L$^{-1}$) for practical salinities $< 32$. Within this salinity range, the two sections could not be clearly distinguished in T-S space, except for station 50, which showed higher potential temperatures at practical salinities between 30 and 31. In contrast, the three sampling regions of the Amundsen Basin (Section 1), the Makarov Basin (Section 2), and the stations north of Greenland (Section 2) differed significantly in $^{129}$I, suggesting differences in the proportions of Atlantic-origin water carrying the tracer signal (see Section 3.4 for further discussion). In general, concentrations were more variable throughout the PML in Section 1, also within the same station (Fig. 5c). The PML in Section 1 extended to a practical salinity of 34 and there was no UHC layer, but we still observed a change in $^{129}$I at a practical salinity of 32.5, approximately coinciding with the upper limit of the UHC layer in Section 2.

The highest $^{129}$I concentrations (around $900 \times 10^7$ at L$^{-1}$) of the entire SAS2021 dataset were found in the Amundsen Basin for practical salinities lower than 34, corresponding to the lower PML in Section 1 (Fig. 5c). For Section 2, the same practical salinity range corresponds to UHC waters and here, an increase in $^{129}$I concentrations with increasing salinity was observed. The highest $^{129}$I concentrations in Section 2 corresponded to the transition between UHC and LHC, at a practical salinity of around 33.5. In contrast to the lower salinity range (PML), in the UHC layer (practical salinities between 32.5 and 34), samples from north of Greenland showed similar $^{129}$I concentrations compared to those from the Makarov Basin. In the LHC (practical salinities $> 34$ but $\sigma_\Theta < 27.7$), the few available samples from the Amundsen (Section 1) and Makarov Basins (Section 2) had similar $^{129}$I concentrations (no samples from the stations north of Greenland were available for $^{129}$I in this salinity range). The low $^{129}$I concentration of station 5 in the Nansen Basin likely reflects the tracer concentration carried by inflowing FSBW (Wefing et al., 2021).

For samples from the Atlantic and intermediate layers in the Atlantic Water core, close to the temperature maximum (practical salinities around 34.8), differences in $^{129}$I between both sections of up to $200 \times 10^7$ at L$^{-1}$ were observed (Fig. 5c). At similar practical salinities, samples from Section 1 showed higher $^{129}$I concentrations, with a clear trend of decreasing $^{129}$I from the Lomonosov Ridge (stations 25 and 26) towards the Nansen Basin (station 5, also observed in Fig. 3). Along Section 2, the highest $^{129}$I concentrations were found close to the Lomonosov Ridge (stations 28 and 30) and decreased towards the north of Greenland. At the Lomonosov Ridge (between station 26 and 28), $^{129}$I in the AAW decreased from the Amundsen to the Makarov Basin, which was also well observed in the profiles at depths between 500 and 1000 m (Fig. 3a and c). The lowest $^{129}$I concentrations in AAW within Section 2 were found in Stations 50 and 52, close to Greenland. For $^{236}$U (Fig. A2), a different trend between Section 1 and Section 2 was apparent. Here, in AAW and below, stations from Section 2 had higher $^{236}$U concentrations compared to stations from Section 1 and the highest $^{236}$U concentration was found in station 46. The distribution of $^{129}$I and $^{236}$U in the Atlantic Water layer will be further discussed in Section 3.4.

## 3.3 Temporal Changes in $^{129}$I Concentration in PSW

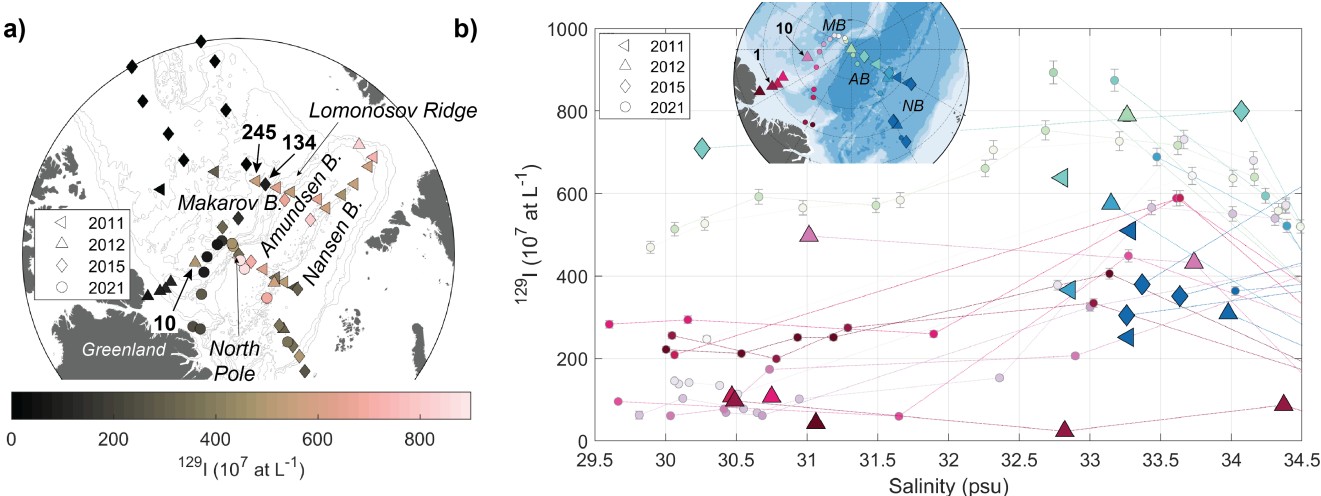

**Figure 6.** (a) Isosurface map of $^{129}$I concentrations in the surface ($< 15$ m depth for 2011, 2015, 2021; 25 m depth for 2012 due to sampling depth resolution). (b) Plot of $^{129}$I concentration against practical salinity for available stations from 2011, 2012, 2015, 2021. The 2021 data is the same as in Fig. 5c. Stations are color-coded by region, following the color-coding for SAS2021 stations.

In all years, the $^{129}$I distribution in surface waters (approximately 5-30 m, depending on sample availability) was characterized by higher concentrations in the Amundsen Basin compared to the Makarov Basin and north of Greenland (Fig. 6a). Changes were only observed close to the Lomonosov Ridge. While the surface Makarov Basin (between the North Pole and Greenland) had low $^{129}$I concentrations ($0-200 \times 10^7$ at L$^{-1}$) in 2021, one station (station 10) suggested a significantly higher concentration of around $500 \times 10^7$ at L$^{-1}$ in this region in 2012. A similar trend was observed further upstream comparing $^{129}$I data from 2011 and 2015: Concentrations decreased from around $550 \times 10^7$ at L$^{-1}$ (2011, Station 245) to around $100 \times 10^7$ at L$^{-1}$ (2015, Station 134).

The sampling resolution for $^{129}$I was sparse, especially in 2011 and 2012, and often limited to "surface samples", with depths varying between 5 and 30 m. The high-resolution $^{129}$I sampling in the upper 50 m at several SAS2021 stations showed significant variation in the $^{129}$I concentration in this highly dynamic layer (Fig. 3a, c and 5c). To account for differences in sampling depths and water mass composition, all samples collected in the vicinity of the SAS2021 stations in different years were added to the SAS2021 $^{129}$I-Salinity plot (Fig. 6b). The available depth profile from the Lincoln Sea in 2012 (station 1), upstream of the SAS2021 stations north of Greenland, showed $^{129}$I concentrations below $100 \times 10^7$ at L$^{-1}$ across the entire Polar Surface Water layer (and below). Higher $^{129}$I concentrations of $200-300 \times 10^7$ at L$^{-1}$ were observed in the SAS2021 stations located further downstream, north of Greenland (dark red circles). As also seen in the isosurface map (Fig. 6a), station 10 from 2012 showed higher $^{129}$I concentrations compared to 2021 samples from nearby stations and corresponding practical salinities. $^{129}$I

concentrations measured at this location in the Makarov Basin in 2012 were close to those observed on top of the Lomonosov Ridge, close to the North Pole, in 2021 (light green circles in Fig. 6b).

## 3.4 Tracer Ages, Dilution, and Mixing of Atlantic Water

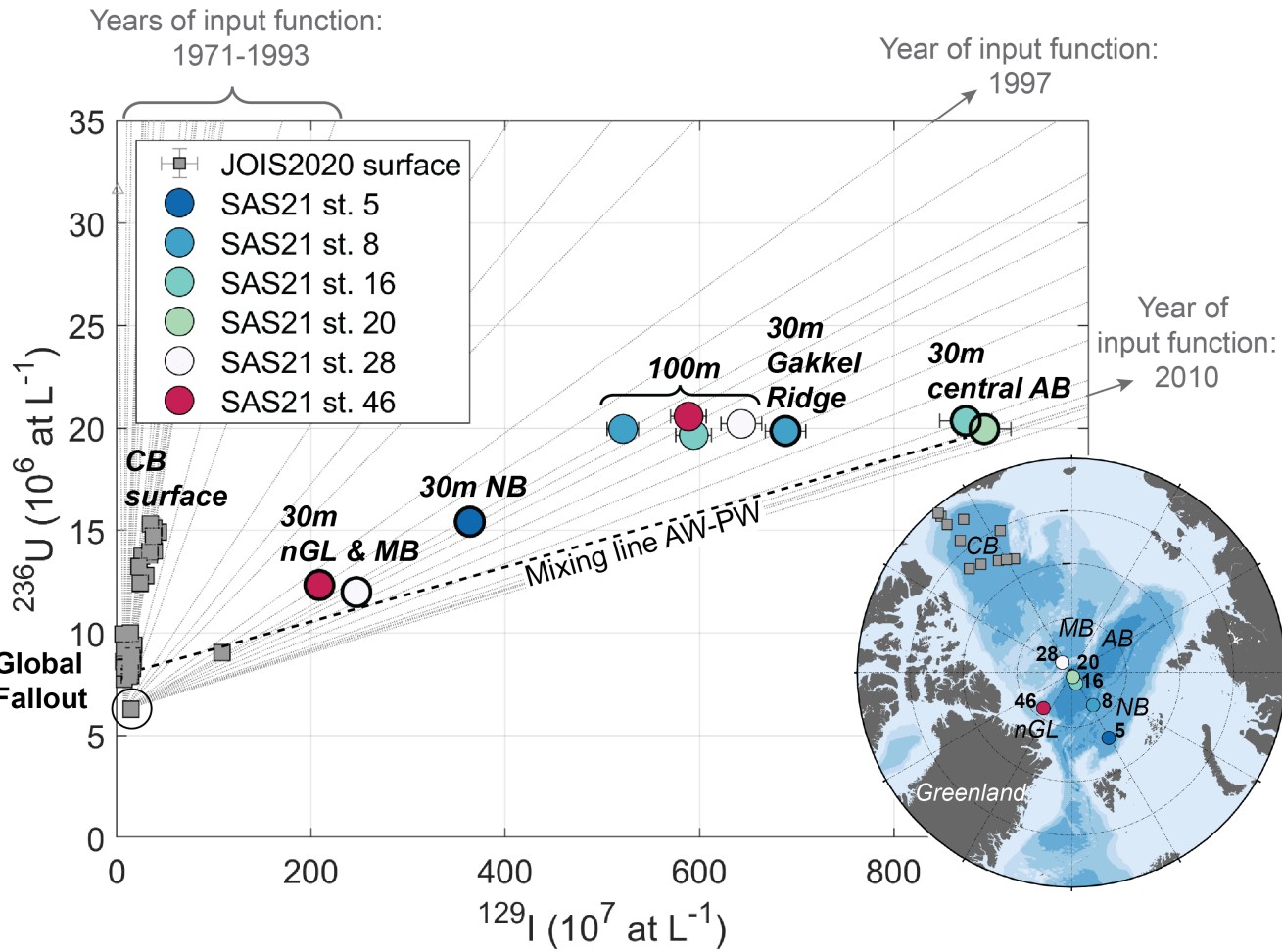

**Figure 7.** Plot of $^{236}$U against $^{129}$I concentrations with binary mixing lines (in grey) between individual years (1997-2010) of the surface layer input function (Fig. 2) and the global fallout background signal. SAS2021 PSW samples are color-coded by station. Thick black outline marks samples from around 30 m depth, thin black outline marks samples from around 100 m depth. CB: Canada Basin, nGL: north of Greenland, MB: Makarov Basin, NB: Nansen Basin, AB: Amundsen Basin. Samples from PSW in the Canada Basin collected in 2020 (JOIS2020, Payne et al., 2024) are shown in grey squares.

In the binary mixing model, PSW samples from SAS2021 plotted on mixing lines from years 1997 to 2010 of the surface layer input function with global fallout, translating to tracer ages of 11 to 24 years (Fig. 7). PSW samples from the Canada Basin (JOIS2020, grey squares in Fig. 7) are shown for comparison. They had much higher tracer ages (30-50 years) and the tracer signal was more diluted compared to the SAS2021 samples. Within SAS2021, surface waters (around 30 m depth, thick black outline) from the central Amundsen Basin (stations 16 and 20) were the youngest (11-17 years). They showed the highest tracer concentrations, i.e., the least dilution with the global fallout signal. The surface waters at station 8 (at the Gakkel Ridge) were older (around 18 years) and more diluted compared to the central Amundsen Basin. The sample from a similar depth at station 5 in the Nansen Basin was older and more diluted than the one from the Gakkel Ridge. Out of all PSW samples, those from the Makarov Basin and north of Greenland (stations 28 and 46) showed the highest dilution of the input function signal and the sample north of Greenland had the highest tracer age of 23 years.

The four available data points from 100 m depth (thin black outline in Fig. 7) showed little variability in tracer concentrations, particularly in $^{236}$U. Their tracer ages were around 18-22 years. The Eurasian Basin samples (station 8 and 16) displayed slightly more dilution of the input function signal compared to the shallower samples from the same stations, whereas the Makarov Basin and north of Greenland samples (station 28 and 46) were substantially less diluted than those from shallower depths.

In AAW, samples from the Makarov Basin and north of Greenland had lower $^{129}$I concentrations paired with higher $^{236}$U concentrations compared to samples from the Eurasian Basin (Fig. A3). AAW samples from station 46 north of Greenland plotted on a mixing line between AAW from the Eurasian Basin and AAW from the Canada Basin (JOIS2020, Payne et al., 2024). For AAW, no tracer ages were derived from the binary mixing model. Instead, transport times were obtained from the TTD model.

### 3.5 Transport Times and Mixing in the Atlantic Layer from the TTD Model

The TTD parameters $\Gamma$ and $\Delta$, and the derived $\Delta/\Gamma$ ratio and $t_{mode}$ for SAS2021 and JOIS 2020 (Canada Basin) are presented in Fig. 8. For SAS2021, mean ages were overall in the range of 15-55 years, $\Delta$ largely between 9-40 years, $\Delta/\Gamma$ ratios were in the range of 0.4-1.2, and mode ages were below 30 years. Mean and mode ages both decreased from the Gakkel Ridge (station 8) towards the Lomonosov Ridge (station 20, Fig. 8a and d). From there, both ages increased towards the Makarov Basin and north of Greenland. Both trends were more pronounced in the mode age compared to the mean age, especially the abrupt change in the mode age when crossing the Lomonosov Ridge (from station 20 to 28). $\Delta/\Gamma$ ratios were generally higher in the Amundsen Basin (between 0.4 and 1.2) compared to the Makarov Basin and north of Greenland (both between 0.4 and 0.6). $\Delta/\Gamma$ ratios were lower in the Canada Basin than in all other basins. The Nansen Basin (station 5) showed a large spread in $\Gamma$ and $\Delta$.

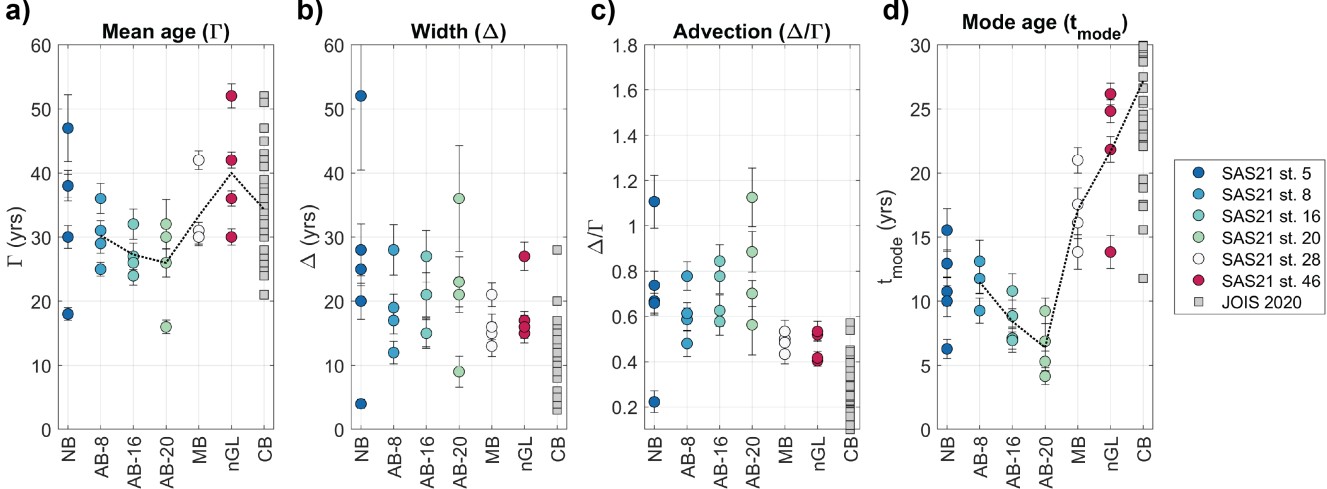

**Figure 8.** TTD parameters $\Gamma$ (mean age, a), $\Delta$ (width, b), derived $\Delta/\Gamma$ ratio (advection, c), and derived $t_{mode}$ (mode age, d) plotted against different Arctic Basins for Atlantic layer samples from SAS2021 (down to 1000 m depth), color-coded by station. NB: Nansen Basin, AB: Amundsen Basin, MB: Makarov Basin, nGL: north of Greenland, CB: Canada Basin. Samples from the Atlantic layer in the Canada Basin collected in 2020 (JOIS2020, Payne et al., 2024) are shown in grey. Dashed lines in (a) and (d) connect mean values for each station.

Tracer concentrations north of Greenland pointed to a mixture of Atlantic Water coming from the Canada Basin and the Amundsen Basin (Fig. A3). We assumed that Atlantic Water from each available sampling depth at station 46 can be described as a mixture of waters from similar depths at station PP7 in the Canada Basin and station 20 in the Amundsen Basin (Fig. 9a).

This was corroborated both by a T-S plot (Fig. 9b), as well as a $^{236}$U-$^{129}$I plot (Fig. 9c), where dotted lines indicate linear mixing of CB and AB waters from respective depths. Samples from station 46 (dark red) plotted close to or directly on the mixing lines. Based on the mixing lines in $^{236}$U-$^{129}$I tracer space, we calculated fractions of Amundsen Basin water found at each depth of station 46, and from those, the bimodal TTD from the individual TTDs of the end-members assuming a linear combination of two IG-TTDs (Eq. (4)).

The resulting bimodal PDFs for station 46 are shown in Fig. 9d (dark red, dashed), along with the unimodal PDFs of both end-members and station 46 (dark red, continuous). Compared to the PDFs from the Canada and Amundsen Basins (gray and green, respectively, in Fig. 9d), the unimodal PDFs at station 46 had a large width (large $\Delta$) and hence a lower maximum probability, indicating the presence of more different ages within the distribution. Mean ages derived for station 46 from the

405 bimodal TTD as a linear combination of the mean ages from the constituents were lower compared to those from the unimodal TTD for all depths except 300 m. Furthermore, the bimodal PDFs decreased more rapidly towards old ages compared to the unimodal PDFs (Fig. 9d). In general, compared to the unimodal TTD for station 46, the $\Delta$ of the mixture was higher for the upper two samples and lower for the others. When the flow field is described by a bimodal TTD, a single mode age cannot be attributed. Instead, the ages with the local maxima in the PDF describe the influence of the two end-members. At station 46,

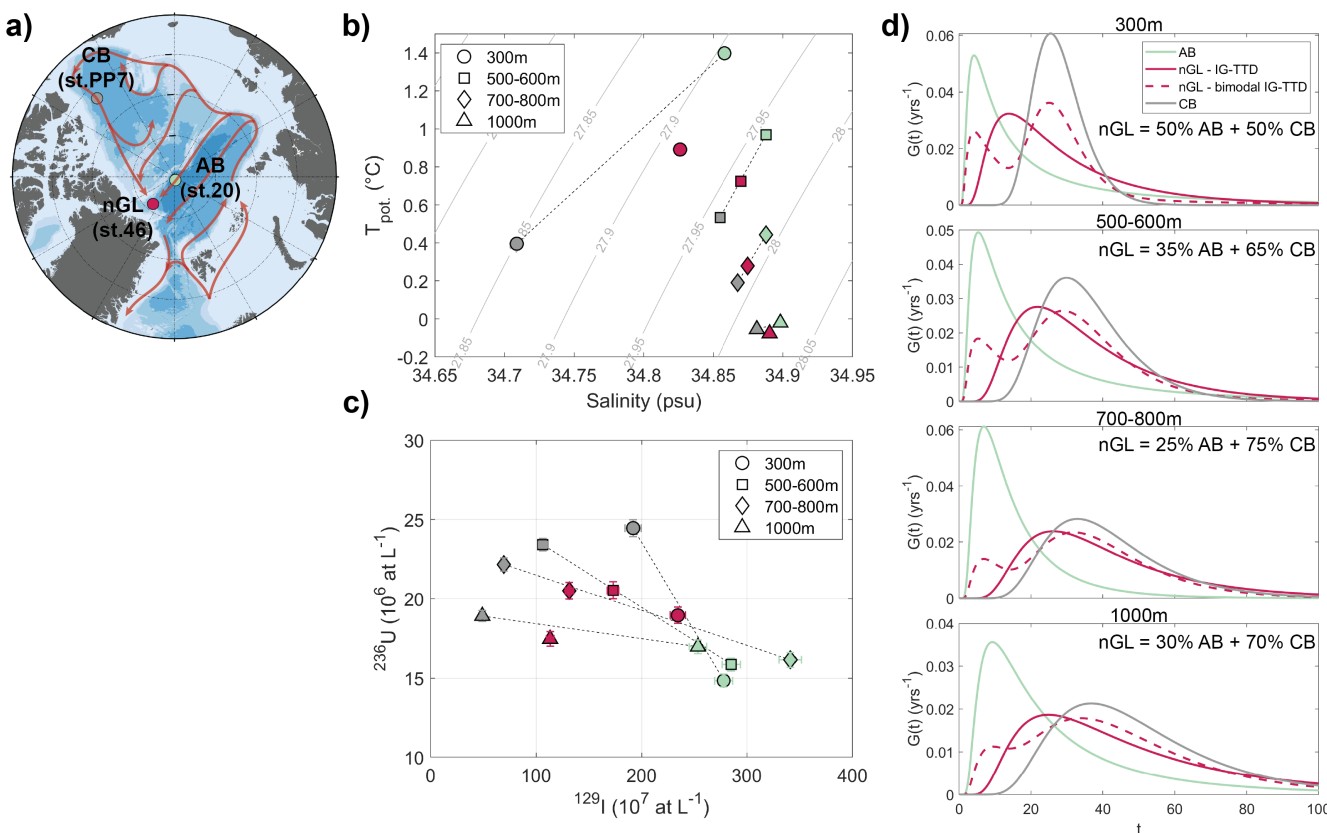

**Figure 9.** (a) Map showing station PP7 in the Canada Basin (CB, Payne et al. (2024)), station 20 in the Amundsen Basin (AB) and station 46 north of Greenland (nGL). Red arrows depict suggested Atlantic Water circulation, where station 46 is a mixture of waters from the other two stations. (b) T-S plot (potential temperature against practical salinity) of samples from available depths (300-1000 m, different symbols) from all three stations (color-coded). (c) $^{236}$U-$^{129}$I plot of samples from available depths (different symbols) from all three stations (color-coded). (d) PDFs G(t) for available depths, color-coded by station. Dashed line depicts the bimodal IG-TTD for station 46.

the overall highest probability in the PDF was associated with the Canada Basin branch for all depths.

## 3.6    Temporal Changes in TTD Parameters in the Atlantic Layer between 2011 and 2021

To assess temporal changes in Atlantic Water transport, we obtained the PDFs from the TTD model with $^{129}$I and $^{236}$U for available stations and depths within different Arctic regions, spanning the years 2011 to 2021 (Fig. 10, see Table 1 for details

on available tracer data).

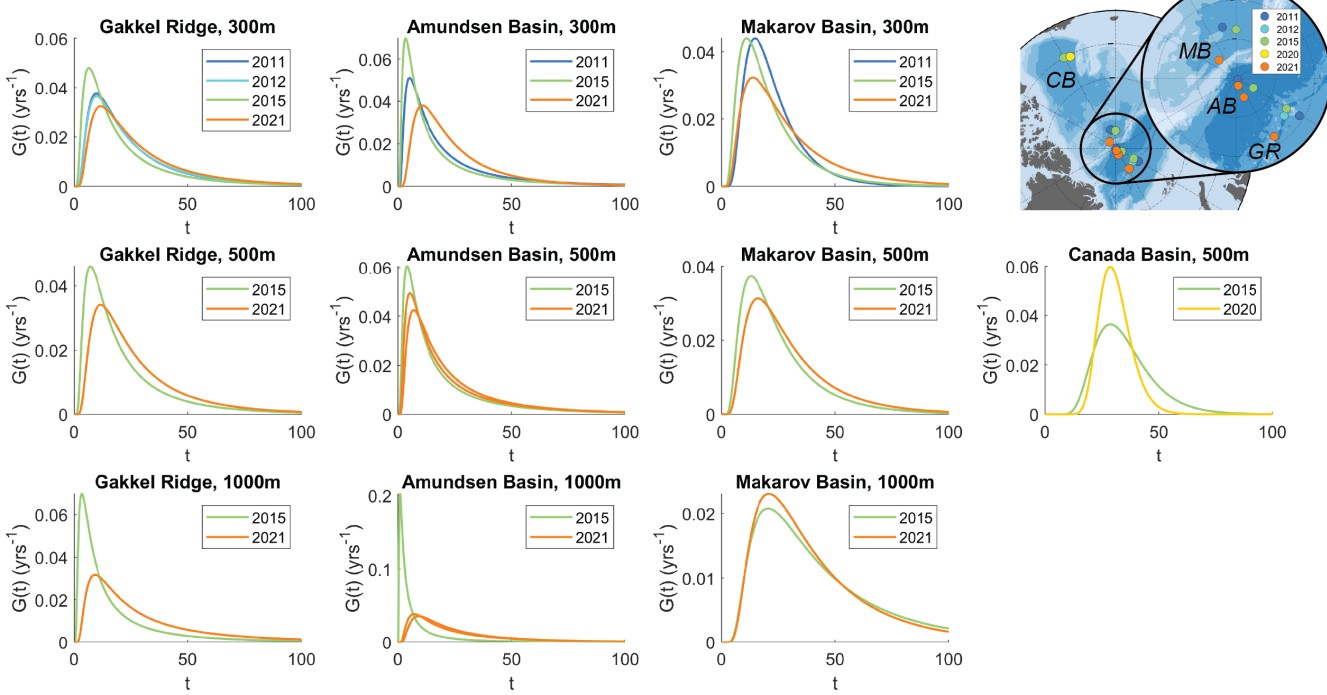

**Figure 10.** PDFs G(t) (different scales) for matching stations from different sampling years (2011 - 2021). Rows are different depths (300, 500, 1000 m), columns are different regions/basins. Stations and regions are shown on the map. GR: Gakkel Ridge, AB: Amundsen Basin, MB: Makarov Basin, CB: Canada Basin. TTD parameters can be found in Appendix C.

For the Eurasian Basin (stations at Gakkel Ridge and Amundsen Basin), the PDFs showed an increase of mean and mode ages over time, especially from 2015 to 2021. $\Delta$ did not show a clear trend and was largely similar in all years (Fig. A4 and Appendix C). PDFs from the Makarov Basin showed a similar behavior between 2015 and 2021 down to 500 m depth. As for the Amundsen Basin, $\Delta$ remained largely unchanged over time. In contrast to the Amundsen Basin, mode ages in the Makarov Basin in 2015 and 2021 did not show a clear increasing trend. In the Canada Basin, samples from different years were only available at 500 m depth. While no change in the mode ages was observed, the shape of the PDF was altered due to a decrease in both $\Gamma$ and $\Delta$ from 2015 to 2020 (Fig. A4).

## 4 Discussion

### 4.1 Circulation Timescales in the Surface Layer

The circulation times obtained for Atlantic Water in the surface layer in 2021 are in line with the general circulation scheme (e.g., Rudels, 2009) and with findings from earlier studies based on the same tracers (Smith et al., 2011; Wefing et al., 2021;

Casacuberta and Smith, 2023; Payne et al., 2024). Young waters with high tracer concentrations found in the central Amundsen Basin suggest a more direct transport of waters containing the tracer signal compared to the southern Eurasian Basin, potentially as part of the Transpolar Drift. In the Nansen Basin, lower tracer concentrations likely reflect inflowing FSBW, which carries lower tracer concentrations than the NCC (Casacuberta et al., 2018). The much higher tracer ages (30-50 years) in PSW samples from the Canada Basin (JOIS2020, grey squares in Fig. 7) compared to the SAS2021 samples from the central Arctic are due to the longer travel time of Atlantic Water to the Canada Basin. The higher dilution of the tracer signal in the surface Canada Basin is explained by the fact that mainly Pacific Water is present at the surface. North of Greenland, the higher tracer age compared to the Amundsen Basin suggests a mixture of Amundsen and Canada Basin waters.

The estimates for lateral surface layer circulation times are particularly valuable since gas tracers do not provide reliable estimates due to their exchange with the atmosphere. In this study, the assessment of circulation patterns and corresponding timescales in the surface layer in 2021 is limited by the spatial resolution of sampling stations for $^{129}$I and $^{236}$U, mainly focusing on parts of the Amundsen Basin and the Makarov Basin close to the Lomonosov Ridge. Furthermore, circulation times obtained from $^{129}$I and $^{236}$U depend on a well-constrained input function of both tracers. Future studies targeting the inflow region of Atlantic Water to the Arctic, such as the recent study by Pérez-Tribouillier et al. (2025), will help to improve our understanding of the Atlantic Water pathways. Especially in shelf seas such as the Kara and Laptev Seas, an improved quantification of freshwater input diluting the tracer signal will allow us to disentangle the water masses in the surface layer of the Arctic Ocean further.

## 4.2 $^{129}$I as a Tracer for the Lateral Pacific Water Extent

Since $^{129}$I is introduced via Atlantic-origin waters and diluted by Pacific inflow, its distribution across the SAS2021 section reflects the Atlantic–Pacific water mass composition in PSW (Smith et al., 2011; Casacuberta et al., 2018; Casacuberta and Smith, 2023). Here, we specifically address the lateral extent of Pacific Water, hence the front between Atlantic- and Pacific-derived waters. In order to assess the presence of Pacific water in the study area based on the tracer concentrations, however, other tracer-free water masses have to be considered, here meteoric water (net precipitation and river runoff) and sea-ice meltwater.

The overall decrease in $^{129}$I (and $^{236}$U) with decreasing salinity throughout the PML in Section 1 (Fig. 5c and Fig A2) points to a dilution of the tracer signal with low-salinity waters. A combination of practical salinity, stable oxygen isotopes ($\delta^{18}$O), and nutrient ratios (nitrate to phosphate, referred to as the N:P method) was used to calculate meteoric, sea-ice meltwater, Atlantic and Pacific Water fractions (e.g., Östlund and Hut, 1984; Jones et al., 1998; Bauch et al., 1995, 2011; Paffrath et al., 2021), here using end-member values from Bauch et al. (2011) and following the calculations outlined therein (Fig. A5a-c). Meteoric water fractions were up to 0.12, highest in the upper 50 m in the Amundsen and Makarov Basins. In contrast, sea-ice meltwater fractions were mostly negative (between -0.04 and 0), pointing to brine rejection and net sea-ice formation, and were

highest for the stations north of Greenland (up to 0.02).

$^{129}$I concentrations were then corrected to 100 % saline water by the calculated meteoric water fractions, assuming negligible to no $^{129}$I input by meteoric water (Casacuberta et al., 2016) (Fig. A5d). The small and largely negative sea-ice meltwater fractions were omitted. After correcting for the meteoric water fraction, we still observed a decrease in $^{129}$I towards the surface, showing that dilution with meteoric water only explains part of the decrease in $^{129}$I towards low salinities. Dilution with older Atlantic Water (carrying lower $^{129}$I concentrations, see Fig. 2a) cannot explain the decrease in $^{129}$I concentrations, as this would

be accompanied by an increase in $^{236}$U (Fig. 2b). Therefore, dilution with Pacific Water most likely explains the decrease of $^{129}$I concentrations towards the surface.

Pacific Water fractions have been estimated using various parameters (e.g., silicate, nutrient ratios, gallium), each with its strengths and limitations. These methods remain debated and carry large uncertainties (e.g., Alkire et al., 2015, 2019; Whitmore

et al., 2020), however, they still serve for an overview of the spatial distribution of Pacific Waters across the sampling area. For the SAS2021 samples, N:P-based Pacific Water fractions suggested that Pacific Water was present in all PSW samples, with the highest fractions of 0.8-1 found in the Makarov Basin (Fig. A5c), supporting the hypothesis of tracer dilution by Pacific Water at all stations. This was also corroborated by the dual-tracer plot (Fig. 7), where samples from north of Greenland and the Makarov Basin fell on a mixing line between the central Amundsen Basin samples to Canada Basin surface samples close

to global fallout. From the N:P method, we obtained similar Pacific Water fractions for PSW in the Amundsen Basin and north of Greenland. At the same time, $^{129}$I concentrations were significantly different between the two regions. This might be caused by an overestimation of the N:P-based Pacific Water fractions for the Amundsen Basin (Alkire et al., 2015; Bauch et al., 2011; Newton et al., 2013), similar to the findings of Alkire et al. (2019). They reported a larger extent of Pacific Water towards the Eurasian Basin based on the N:P method compared to other methods and suggested uncertainties of 40 % or more associated

with N:P-based Pacific Water fractions.

Generally, based on the combined $^{129}$I and $^{236}$U data, we can hence conclude that the front between Atlantic and Pacific-derived waters in the surface layer of the central Arctic was located somewhere between station 20 and station 28 in 2021, due to the different location of surface samples from these two stations in the $^{236}$U-$^{129}$I mixing plot (Fig. 7). With the better spatial

resolution for $^{129}$I, and the sharp decrease in surface $^{129}$I concentration between station 26 (on top of the Lomonosov Ridge) and station 28 (Makarov Basin) (Fig. 3 and Fig. 5), we can restrict it further to the Makarov Basin side of the Lomonosov Ridge (Fig. 11a). However, the presence of some Pacific Water is required to explain the lower $^{129}$I concentrations in the surface waters of stations 25 and 26 compared to station 20, which is supported by the N:P method.

For a better assessment of the position of the Atlantic-Pacific-Water front, an increased spatial sampling resolution of at least $^{129}$I would be preferable. With the available data, we can constrain the position across the sampled transect, but not in other regions across the Arctic Ocean. It cannot be concluded from this dataset whether the front was generally aligned with the

Lomonosov Ridge or only happened to be at the ridge at the sampled stations.

## 4.3 Temporal Changes in the Pacific Water Extent

With $^{129}$I serving as a tracer for the lateral extent of Pacific Water, the observed change in $^{129}$I distribution in the central Arctic between 2011 and 2021 is likely driven by a shift in the relative contribution of Atlantic- and Pacific-derived waters. The higher tracer concentrations observed in the surface Makarov Basin close to the Lomonosov Ridge in 2011 and 2012 compared to 2015 and 2021 point to a higher fraction of Atlantic Water in 2011/12. This implies that Pacific Water was reaching further into the central Arctic in 2015/2021 compared to 2011/12 (Fig. 11a, see also Wefing et al. (2021)).

Atlantic and Pacific Water fractions for the same datasets were also calculated based on the N:P method, again following the method by Bauch et al. (2011, see also Section 4.2), albeit using a three-end-member model of Atlantic, Pacific, and freshwater due to limited availability of $\delta^{18}$O data (Appendix B). In 2011, high Atlantic Water fractions were observed in the Makarov Basin, extending to the Alpha-Mendeleyev Ridge (Fig. 11b). In 2015 and 2021, Atlantic Water fractions in the surface were limited to the Eurasian side of the Lomonosov Ridge instead, which is in line with the water mass distribution obtained from $^{129}$I.

In summary, the distribution of $^{129}$I and the N:P-based Atlantic Water fractions suggested that Atlantic-origin water extended farther across the Lomonosov Ridge into the Makarov Basin in 2011 and 2012, compared to the years 2015 and 2021. This implies a shift in the surface water circulation pattern. Smith et al. (2021) found that changes in the surface $^{129}$I distribution across the Arctic Ocean reflected a change from a cyclonic circulation mode associated with a positive AO index in the mid-1990s to an anti-cyclonic circulation mode along with a negative AO index in the 2000s (Fig. A7). This change was accompanied by a shift in the Atlantic-Pacific Water front from the Alpha-Mendeleyev Ridge (1990s) to the Lomonosov Ridge (2015). Our analysis of $^{129}$I concentrations between 2011 and 2021 indicated that the position of the front still changed between 2011 and 2015, but remained stable between 2015 and 2021. Polyakov et al. (2023) described an increasingly positive phase of the Arctic Dipole (AD+) between 2007 and 2021, leading not only to enhanced inflows of Atlantic Water through the Barents Sea Opening compared to Fram Strait, but also to a shift in the alignment of the Transpolar Drift from the Amerasian Basin towards the Lomonosov Ridge. This is further supported by the change in surface $^{129}$I distribution, where the available tracer data suggests that a strong shift occurred between 2011 and 2015.

## 4.4 Provenance of Halocline Waters

Below the surface layer, which was found to be clearly influenced by Pacific Water, halocline waters reside. Based on their T and S properties, halocline layer samples from the Amundsen Basin (station 8 and 16) were associated with the LHC, which in the Eurasian Basin is formed through repeated winter convection cycles north of the Barents Sea, as well as by advection of

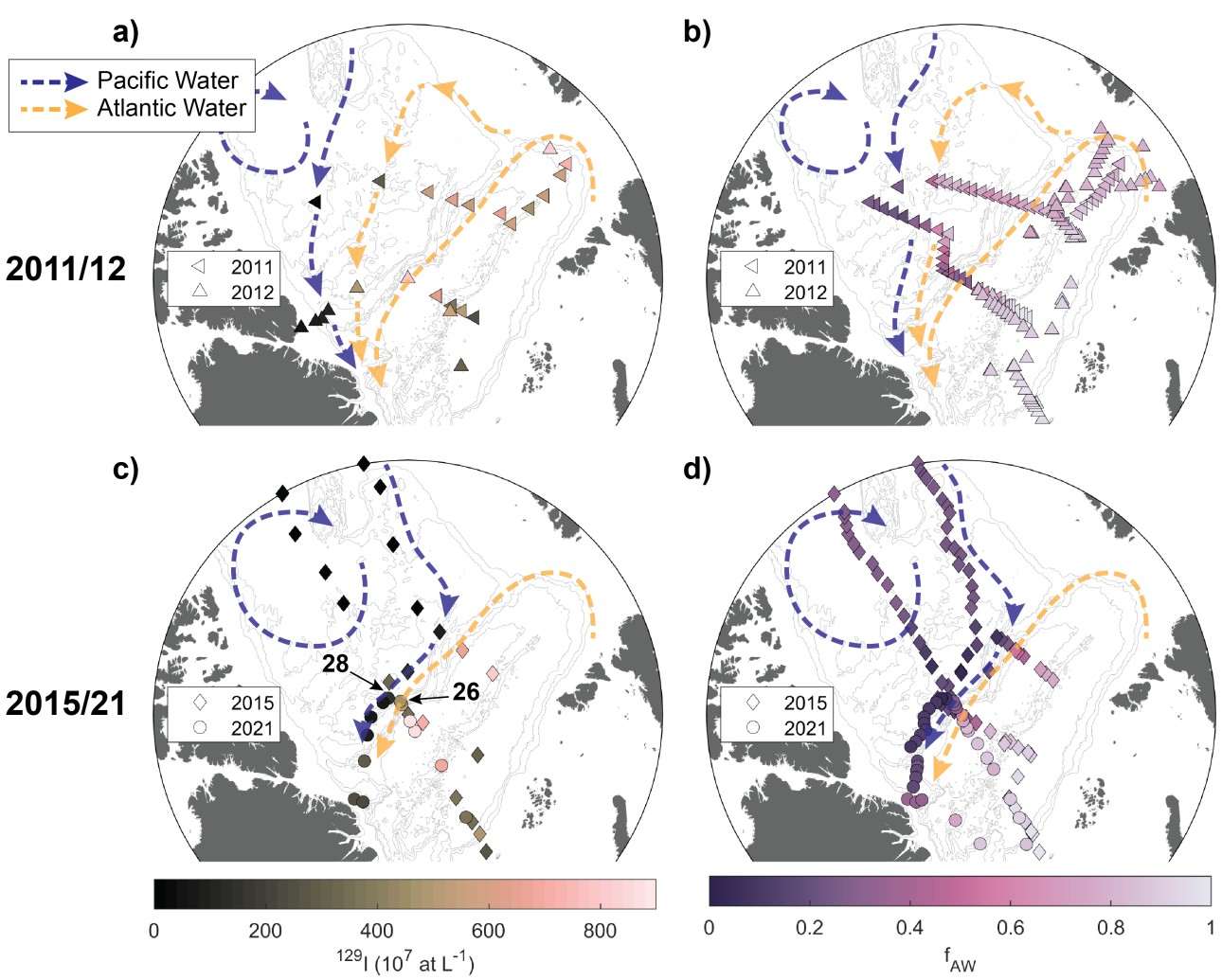

**Figure 11.** (a) Isosurface map of $^{129}$I concentrations in the surface in 2011 and 2012 ($< 15$ m depth for 2011 25 m depth for 2012 due to sampling depth resolution). (b) Isosurface map of Atlantic Water fractions ($f_{AW}$) derived from practical salinity and N:P ratios, for all available data from the surface layer from the same expeditions as in (a). (c) Isosurface map of $^{129}$I concentrations in the surface in 2015/2021 ($< 15$ m depth). Stations 26 and 28 from SAS2021 are highlighted. (d) Isosurface map of Atlantic Water fractions ($f_{AW}$) derived from practical salinity and N:P ratios, for all available data from the surface layer from the same expeditions as in (c). In all panels, the circulation of Atlantic- and Pacific-derived surface waters (yellow and blue, respectively) is sketched based on the corresponding tracer distribution (a & c) or derived water mass fraction (b & d).

low-salinity shelf water, primarily from the Laptev Sea (Rudels et al., 1996). The halocline samples from the Makarov Basin (station 28) and north of Greenland (station 46) had practical salinities slightly below 34. They were, therefore, classified as UHC waters by hydrographic properties and are thought to be derived from Pacific inflow but also influenced by the East Siberian Sea (e.g., Anderson et al., 1994, 2017). However, this is not supported by the radionuclide concentrations. The similar location of the 100 m samples from the different stations in the dual-tracer plot (Fig. 7) suggested a common origin or formation region and transport route for these halocline waters, before they are transported across the Arctic Ocean towards Greenland. This was also supported by the additional available $^{129}$I data, where samples from the Amundsen and Makarov Basins showed similar $^{129}$I concentrations throughout the LHC layer (Fig. 5c and Fig. A6a). Furthermore, the comparably high tracer concentrations, substantially higher than closer to the surface, suggested only little Pacific-origin water but a significant fraction of tracer-labeled NCC-origin waters, which were transported along the Eurasian shelf and contributed to the formation of halocline waters along their way. Based on $^{129}$I and $^{236}$U, we conclude that halocline waters formed from Atlantic-origin waters were present throughout the entire study area, including the Makarov Basin and the area north of Greenland.

The distribution of the NO parameter (NO = $(9 \times NO_3^-) + O_2$, Broecker (1974)) supports this finding. Alkire et al. (2019) used an NO concentration of $400 \, \mathrm{mmol \, m^{-3}}$ to discriminate Atlantic from Pacific waters, with lower concentrations indicative of Atlantic Water. The distribution of NO derived from the SAS2021 dataset at a practical salinity of 34 did not show large variability across the study area (Fig. A6b). NO concentrations in the Amundsen and Makarov Basins and north of Greenland were all below $400 \, \mathrm{mmol \, m^{-3}}$, hence pointing to Atlantic origin.

## 4.5 Circulation of Waters in the Atlantic Layer

As for the surface layer, the analysis of circulation times in 2021 is mainly limited to the western Eurasian Basin and to specific sampling depths. We exclude the Nansen Basin from further interpretation since this station is not located downstream of the initialization point of the Atlantic Layer input function (the St. Anna Trough, where FSBW and BSBW merge), and the TTD analysis is not expected to provide meaningful results (Wefing et al., 2021; Raimondi et al., 2024; Pérez-Tribouillier et al., 2025).

The decrease in mode ages from the central Eurasian Basin towards the Lomonosov Ridge suggests that waters in the mid-depth Atlantic layer are transported faster close to the ridge, which could be explained by transport in a branch of the Arctic Ocean Boundary Current flowing along the Amundsen Basin side of the Lomonosov Ridge. The same spatial pattern was observed by Körtke et al. (2024), investigating tracer ages from CFC-12 and SF$_6$, as well as Pasqualini et al. (2024), from $^3$H-$^3$He ages. However, interestingly, $\Delta/\Gamma$ ratios did not suggest very advective transport at station 20, but rather increased from the interior Eurasian Basin towards the Lomonosov Ridge, indicating more mixing along the flow. In the Makarov Basin, north of Greenland, and in the Canada Basin, low $\Delta/\Gamma$ ratios and high mode ages pointed to more advective flow, and longer current pathways, confirming earlier studies on TTDs in the Canada Basin (Smith et al., 2011; Wefing et al., 2021; Raimondi et al.,

2024). The significantly higher mode ages in the Makarov Basin suggest that waters have been transported along a different (longer) loop compared to the Amundsen Basin. A difference in water mass ages between both sides of the Lomonosov Ridge was also observed by Tanhua et al. (2009) and Gerke et al. (2024), albeit in mean ages derived from gas tracers. Both studies found higher mean ages, i.e., a slower ventilation, in the Makarov Basin compared to the Amundsen Basin, with a sharp front over the Lomonosov Ridge. This is in line with our findings based on $^{129}$I and $^{236}$U, confirming the study by Raimondi et al.

(2024) that anthropogenic radionuclides and gas tracers such as CFCs and $SF_6$ act as similar Atlantic Water tracers in the Arctic Ocean once the Atlantic Water layer is isolated from atmospheric gas exchange.

The binary mixing model and the large width of the unimodal TTD indicated that AAW north of Greenland represents a mixture of Atlantic Water that has circulated in the boundary current through the "long loop" (Canada Basin) and the "short

loop" (Eurasian Basin) and hence a wide distribution of circulation times. The mean ages derived from the bimodal TTD, accounting for two main pathways and associated circulation times, were more reasonable than those from the unimodal TTD, which on average even exceeded those from the Canada Basin (Fig. 8a). Also the more rapid decrease of the bimodal TTD at old ages (Fig. 9d), similar to the Canada and Amundsen Basin, points to the use of the bimodal TTD to describe Atlantic Water transport north of Greenland which considers the two branches of Atlantic Water being advected from the Amundsen

and Canada Basins. This applies to other regions of the Arctic Ocean as well, such as the mixing of Atlantic and Pacific-derived water over the Chukchi Sea shelf described in Smith et al. (2022).

## 4.6    Temporal Changes in Atlantic Water Circulation

Mode ages are a good measure of lateral circulation times of Atlantic Water (Smith et al., 2011; Wefing et al., 2021; Raimondi

et al., 2024) and the observed increase in mode ages in the Amundsen and Canada Basins between 2015 and 2021 (Fig. A4 and Appendix C) suggests a slowdown in Atlantic Water circulation of about $3 - 8$ years over this time frame, depending on locations and depth. Over the same period, little to no changes in mixing within the Atlantic Water transport were observed (TTD parameter $\Delta$) in different locations in the Amundsen Basin, the Makarov Basin close to the Lomonosov Ridge, and a location in the Canada Basin. Our findings confirm the results by Gerke et al. (2024), who observed a similar trend based on

CFC-12 and $SF_6$ data from SAS2021 and earlier years (1991, 2005, 2015). They found increasing mean ages from 2015 to 2021 throughout the entire Atlantic Water layer in the Amundsen Basin, which they attributed to a weakening of the Arctic Ocean Boundary Current, leading to a slowdown in the lateral circulation of Atlantic Water.

A decrease in the strength of the boundary current between the mid-1990s and 2015 has been observed in tracer studies,

which was related to a decrease in the AO index (Smith et al., 2021; Körtke et al., 2024). Both studies found an increase in Atlantic Water circulation times over that period. Since 2015, the AO index has largely been positive, based on which a decrease in Atlantic Water circulation times between 2015 and 2021 could be expected. This is not observed in our study region. However, it is unclear how fast circulation in the mid-depth Atlantic Layer responds to changes in atmospheric conditions and

surface layer circulation.


Another reason for changes in mean ages, as indicated by gas tracers and radionuclides, could be circulation changes in the formation region of mid-depth waters, i.e., Fram Strait and the Barents Sea. Polyakov et al. (2023) describe fluctuations in the inflow of Atlantic Water through the Barents Sea Opening compared to Fram Strait. This would likely affect radionuclide tracer concentrations introduced into the mid-depth Atlantic layer through different mixing between the different branches

carrying different tracer concentrations (Casacuberta et al., 2018; Pérez-Tribouillier et al., 2025) and gas tracer input functions depending on surface saturation levels (e.g., Raimondi et al., 2021).

The mechanism driving the observed increase in Atlantic Water circulation times from 2015 to 2021 remains unclear and further research is needed in this regard. This could include the simulation of Atlantic Water tracers in circulation models as

done in Karcher et al. (2012) and Smith et al. (2021). Also, an extended temporal coverage with high spatial sampling resolution across multiple Arctic basins is probably required for the mid-depth layer, since changes occur more slowly than in the surface layer. To better understand the link to atmospheric conditions, observations spanning phases of different atmospheric conditions are required, such as the change in AO index from the 1990s to the 2000s.

## 5 Conclusion and Outlook

In this study, we utilized the combination of the radionuclides $^{129}$I and $^{236}$U to assess Atlantic Water pathways, mixing, and circulation times, as well as the Pacific Water lateral extent in the central Arctic Ocean in 2021 and to study changes in Atlantic Water circulation between 2011 and 2021.

Along the transect of the SAS2021 expedition, we observed a sharp drop in radionuclide concentrations in the surface layer

between the Amundsen and the Makarov Basins, directly at the Lomonosov Ridge. We attributed this to a significant fraction of Pacific Water on the Makarov Basin side, carrying low radionuclide concentrations, which was qualitatively supported by Pacific water fractions calculated with the N:P (nitrate to phosphate) method. Regarding the temporal evolution of the lateral extent of Pacific Water, we found that the Atlantic-Pacific Water front was located further in the Makarov Basin in 2011 and 2012 compared to 2015 and 2021, when it was aligned with the Lomonosov Ridge. The same pattern was observed based on

the N:P method. This supports earlier findings by Smith et al. (2021), who describe the shift of the Atlantic-Pacific Water front from the Alpha-Mendeleyev Ridge in the 1990s to the Lomonosov Ridge in 2015 due to changes in the Arctic Oscillation index. Our data indicates that the shift towards the Lomonosov Ridge continued through 2011 and 2012. By 2015, the front had aligned with the Lomonosov Ridge, and was located at a similar location in 2021.

For the halocline layer, we found similar radionuclide concentrations across the entire transect, indicating a common formation area and transport route of the halocline waters. High tracer concentrations furthermore indicated the presence of

Atlantic-origin waters in the halocline, suggesting that these waters were formed around the Barents Sea, where they picked up the tracer signal of the surface layer input function.

This dataset includes the first available $^{129}$I and $^{236}$U data from the area north of Greenland. In both, the surface layer (except the halocline) and the mid-depth Atlantic layer, tracer concentrations indicated that these waters were a mixture of Canada Basin and Amundsen Basin waters. In the mid-depth Atlantic layer, the TTD model provided mean ages, mode ages, and mixing properties for the flow of Atlantic Water through the Arctic Ocean. Confirming earlier studies based on gas tracers (CFCs and SF$_6$), we found an increase in mean (and mode) ages when crossing the Lomonosov Ridge from the Amundsen into the

Makarov Basin, implying a longer transport route. Regarding temporal changes, we observed an increase in mean and mode ages in the Amundsen Basin from 2015 to 2021. This implies either a slowdown in Atlantic Water circulation, which could be attributed to a decrease in the strength of the Arctic Ocean Boundary Current (as also suggested by Gerke et al., 2024), different Atlantic Water pathways (as discussed in Smith et al., 2021; Körtke et al., 2024), or changes in the inflow region of the tracers (such as changes in Fram Strait and Barents Sea Water inflow as a consequence of the Arctic Dipole state, Polyakov

et al., 2023). Further research is needed to identify the mechanism behind these changes.

    The atlantification of the Arctic Ocean is no longer restricted to the Eurasian Basin, as recent findings indicate that it has extended into the Makarov Basin (Polyakov et al., 2025), once again confirming that the Arctic Ocean is undergoing drastic changes. How these changes will affect the Arctic system, from the uptake of anthropogenic carbon to consequences for

ecological systems, remains to be investigated. As part of this, understanding and monitoring changes in the Atlantic Water circulation is crucial. Our study confirms that the anthropogenic radionuclides $^{129}$I and $^{236}$U are effective tracers of Atlantic Water, and highlights the value of the tracer dataset collected over the past decade for assessing temporal changes. During the SAS2021 expedition, an effort was made to couple the sampling of different circulation and ventilation tracers. Following the recent study by Raimondi et al. (2024), this set of various tracers can be combined in future studies, allowing, for instance, a

more comprehensive assessment of anthropogenic carbon stored in the Arctic Ocean. To this end, long-term ventilation tracers such as carbon-14 and argon-39 should be included and will allow extending such studies to deeper and older water masses.

*Data availability.* Radionuclide data ($^{129}$I and $^{236}$U) from the SAS-Oden 2021 expedition are available on Zenodo (Wefing, 2025). Hydrographic data from the same expedition are available on PANGAEA: https://doi.org/10.1594/PANGAEA.951266 and https://doi.org/10. 1594/PANGAEA.951264 (downloaded 2024-08-23). Biogeochemical bottle data are available on GLODAPv2.2023 (Lauvset et al., 2024):

https://glodap.info/index.php/merged-and-adjusted-data-product-v2-2023/ (downloaded 2024-06-17).

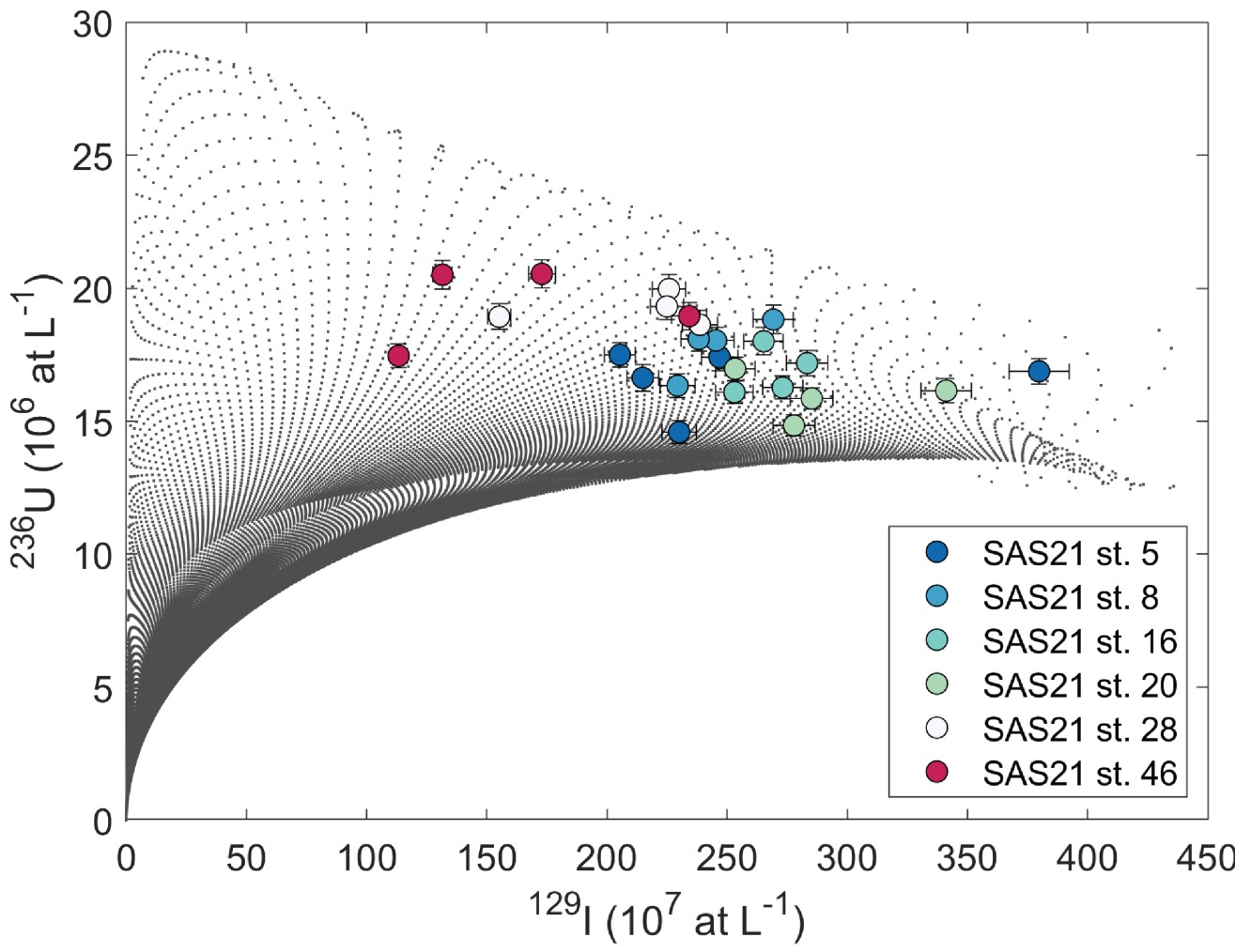

**Figure A1.** $\Delta - \Gamma$ grid in $^{236}$U-$^{129}$I space for the determination of TTD parameters according to "Smith's method" (see Raimondi et al. (2024) for detailed explanation). SAS2021 samples are plotted on top, color-coded by station.

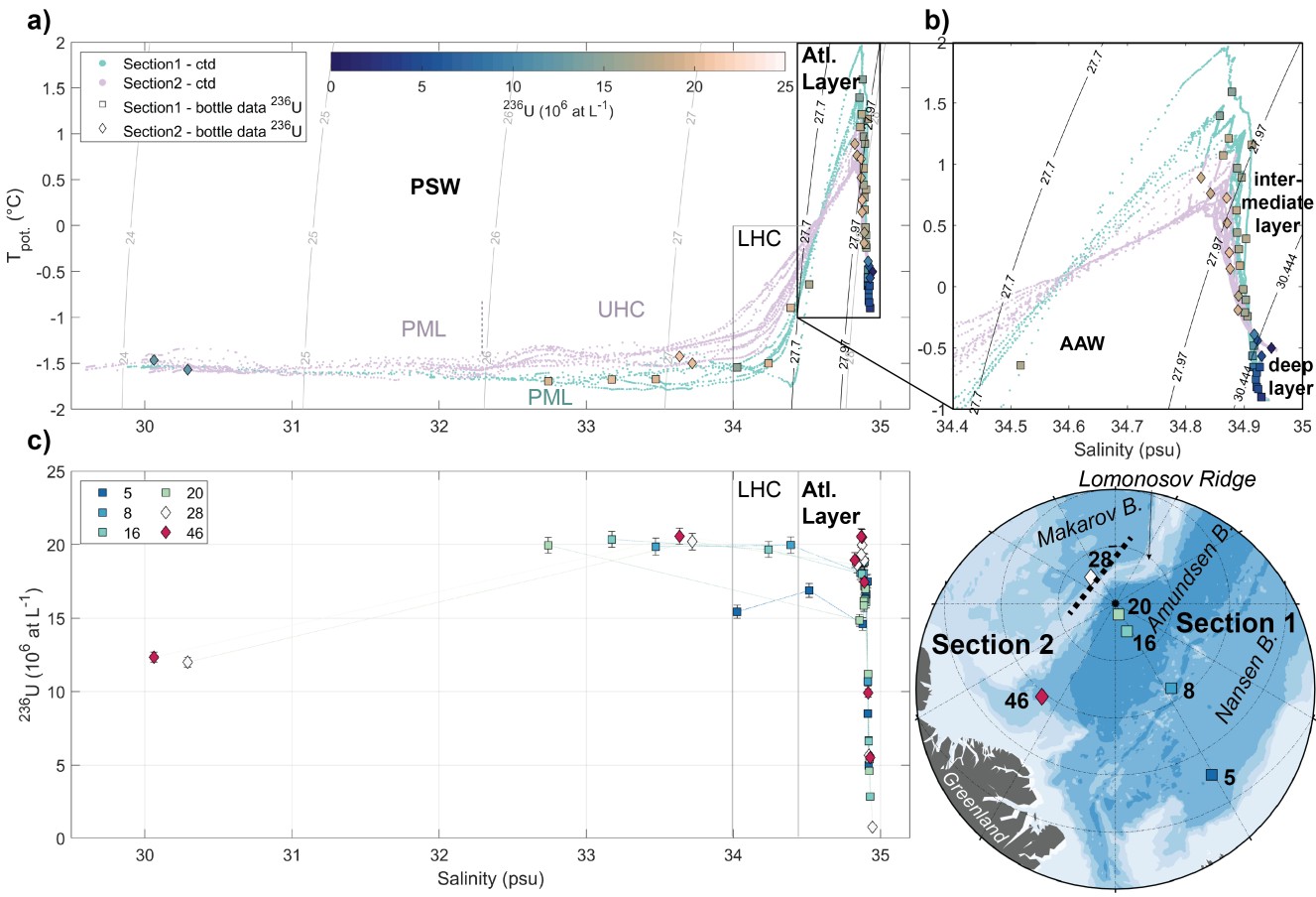

**Figure A2.** (a) T-S plot (potential temperature against practical salinity) with [236]U concentrations color-coded and different water masses indicated (PSW: Polar Surface Water, PML: Polar Mixed Layer, UHC: upper halocline, LHC: lower halocline, AAW: Arctic Atlantic Water). Teal and purple dots are 1 m-binned CTD data from Section 1 and 2, respectively. Square and diamond data points are bottle file data from Section 1 and 2, respectively. (b) close-up of the Atlantic Layer (black box in a). (c) Plot of [236]U concentration against practical salinity with stations color-coded.

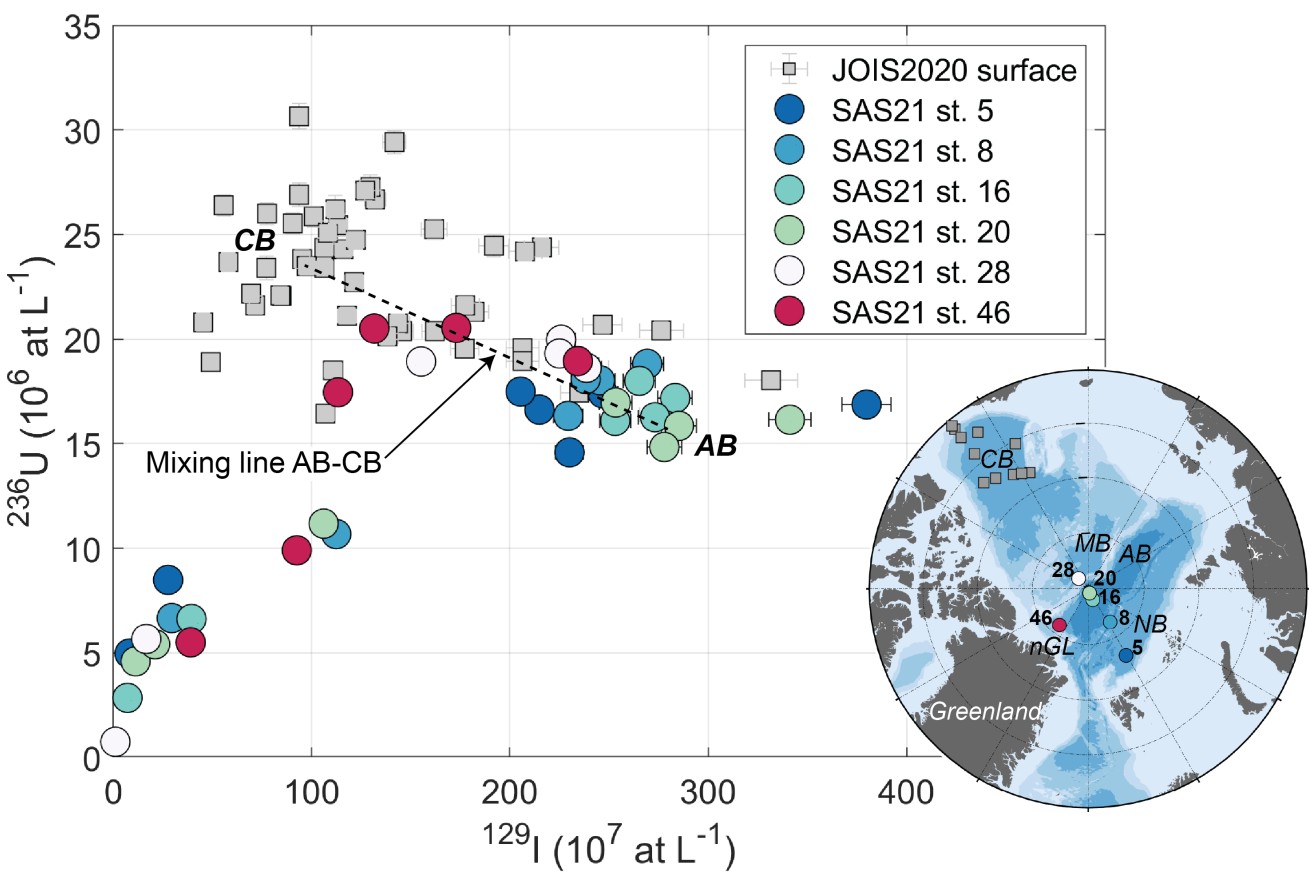

**Figure A3.** Plot of $^{236}$U against $^{129}$I concentrations for samples from the Atlantic Layer. SAS2021 PSW samples are color-coded by station. CB: Canada Basin, AB: Amundsen Basin. Samples from the Atlantic Layer in the Canada Basin collected in 2020 (JOIS2020, Payne et al., 2024) are shown in grey squares.

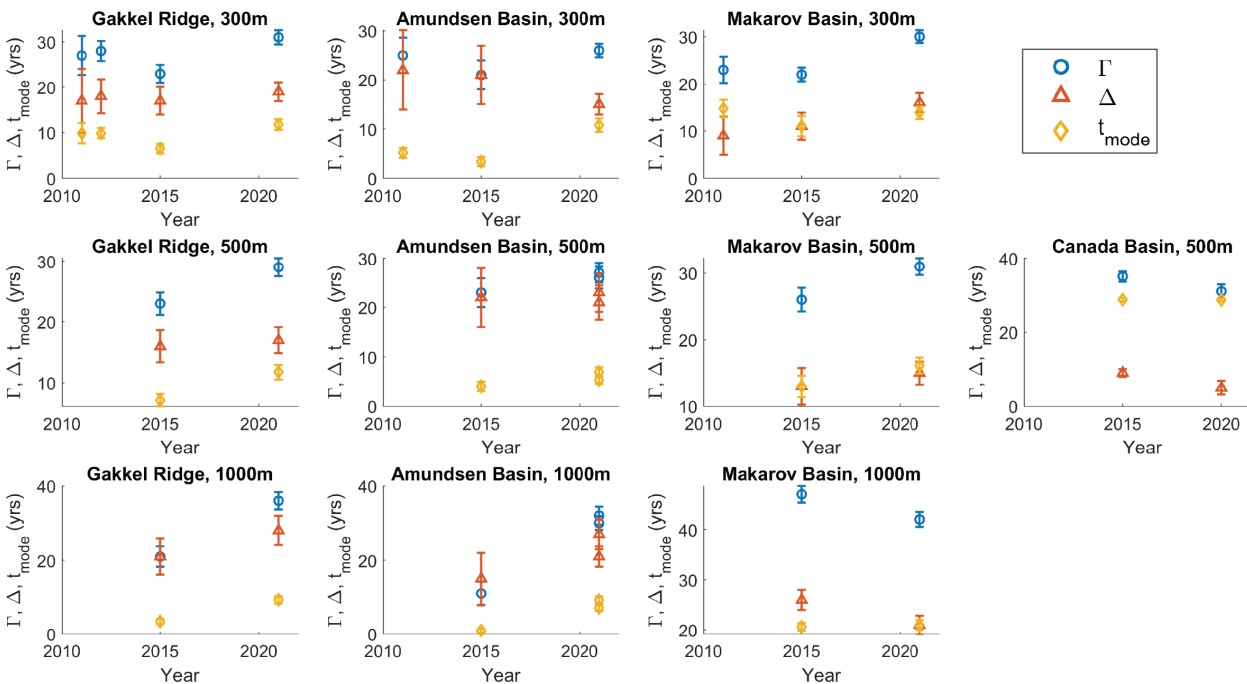

**Figure A4.** TTD parameters $\Gamma$, $\Delta$, and $t_{mode}$ plotted against sampling year. Rows are different depths (300, 500, 1000 m), columns are different regions/basins. Stations and regions are shown in the map in Fig. 10. Data can be found in Appendix C.

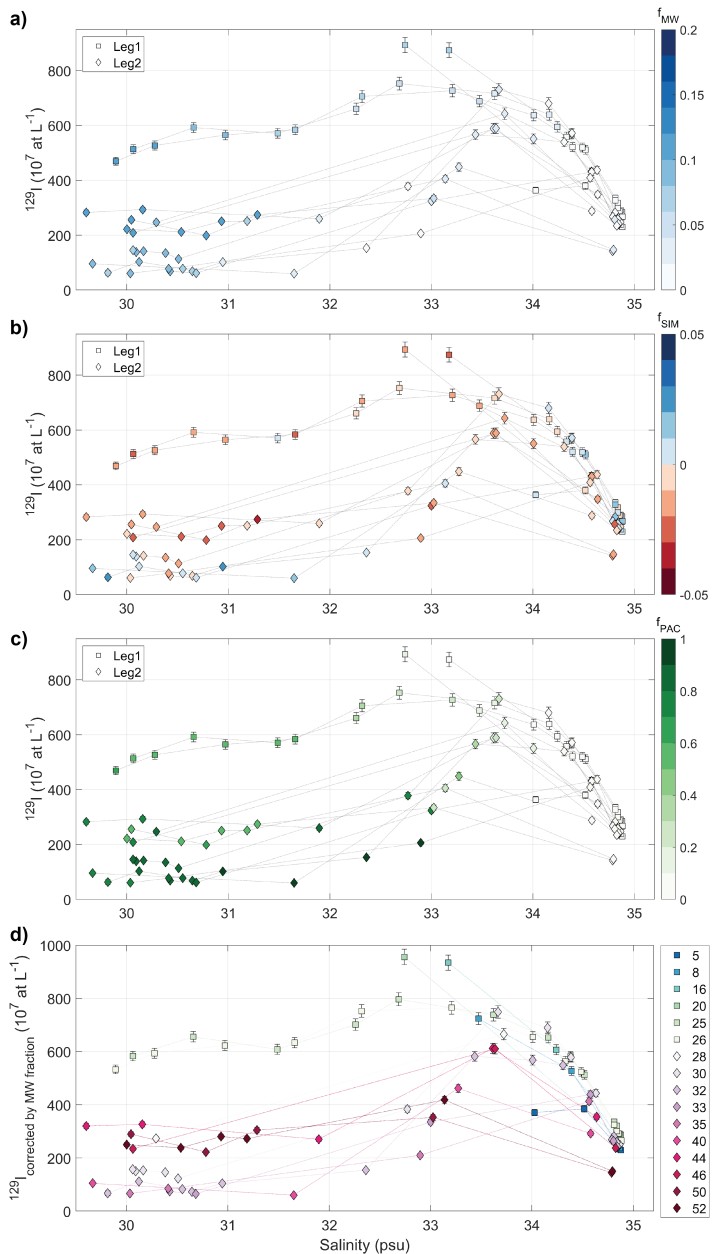

**Figure A5.** Plot of $^{129}$I concentration against practical salinity with data color-coded by (a) meteoric water fraction $f_{MW}$, (b) sea-ice meltwater fraction $f_{SIM}$, (c) Pacific Water fraction $f_{PAC}$. (d) Plot of $^{129}$I concentration corrected by meteoric water fraction $f_{MW}$ against practical salinity with stations color-coded.

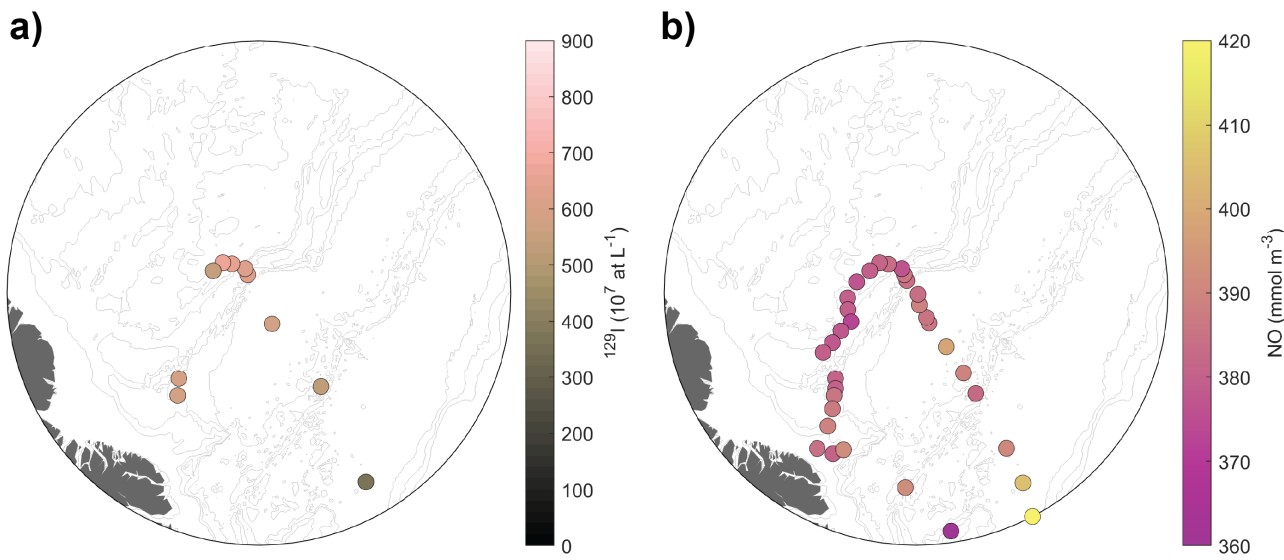

**Figure A6.** Isosurface maps of (a) [129]I concentration and (b) the NO parameter (Broecker, 1974; Alkire et al., 2019) at practical salinity $S_P = 34 \pm 0.5$.

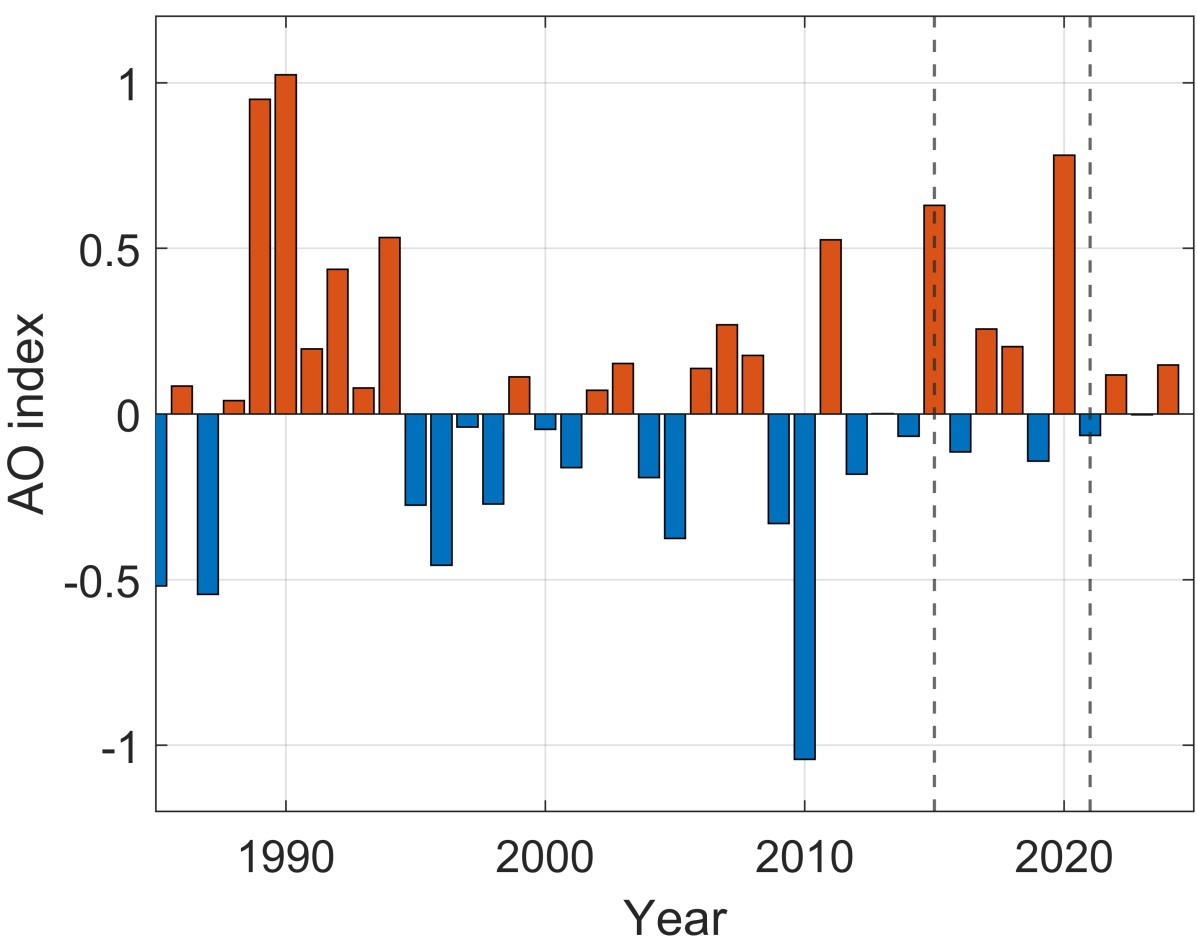

**Figure A7.** Arctic Oscillation (AO) index (yearly average). Tracer sampling years 2015 and 2021 are indicated as dashed lines. Figure modified after Fig. 6d in Körtke et al. (2024). Arctic Oscillation Data downloaded from: https://www.cpc.ncep.noaa.gov/products/precip/ CWlink/daily_ao_index/ao.shtml (28.02.2025).

## Appendix B: Calculation of Atlantic and Pacific Water Fractions with Three-End-Member Model

Atlantic and Pacific Water fractions were calculated based on the N:P method, following the method by Bauch et al. (2011), but using a three-end-member model of Atlantic Water (AW), Pacific Water (PW), and freshwater (FW) due to limited availability of $\delta^{18}O$ data. Endmember values were taken from Bauch et al. (2011), for the freshwater endmember we used the river water endmember therein. The three-component mass balance is the following:

$$f_{AW} + f_{PW} + f_{FW} = 1 \tag{B1}$$

$$f_{AW}S_{AW} + f_{PW}S_{PW} + f_{FW}S_{FW} = S_{meas} \tag{B2}$$

$$f_{AW}P_{AW} + f_{PW}P_{PW} + f_{FW}P_{FW} = P_{meas} \tag{B3}$$

with $S_{AW} = 34.92$, $S_{PW} = 32.7$, $S_{FW} = 0$, $P_{AW} = 0.0596 \cdot N_{meas} + 0.1139$, $P_{PW} = 0.0653 \cdot N_{meas} + 0.9400$, $P_{FW} = 0.1$. $f_{AW}$, $f_{PW}$, and $f_{FW}$ represent the fractions of Atlantic Water, Pacific Water, and freshwater, respectively.

Hydrographic and nutrient data for expeditions PS78 (2011), PS80 (2012), and PS94 (2015) can be found on PANGAEA: https://doi.org/10.1594/PANGAEA.832164, https://doi.org/10.1594/PANGAEA.774181, https://doi.org/10.1594/PANGAEA.834081, https://doi.org/10.1594/PANGAEA.819452, https://doi.org/10.1594/PANGAEA.868396, https://doi.org/10.1594/PANGAEA.859559. Hydrographic and nutrient data for expedition HLY1502 (2015) was taken from the GEOTRACES Intermediate Data Product: https://doi.org/10.5285/cf2d9ba9-d51d-3b7c-e053-8486abc0f5fd.

## Appendix C:  TTD Temporal Results

Tables with TTD parameters $\Gamma$, $\Delta$, and $t_{mode}$ for the different regions/basins, depths (300, 500, 1000 m), and sampling years shown in Fig. A4.

**Table C1.** Gakkel Ridge

| Depth (m) | TTD parameter (years) | 2011 204 val | unc | 2012 378 val | unc | 2015 68 val | unc | 2020 – val | unc | 2021 8 val | unc |
|---|---|---|---|---|---|---|---|---|---|---|---|
| 300 | $\Gamma$ | 27 | 4 | 28 | 2 | 23 | 2 | NaN | NaN | 31 | 2 |
| | $\Delta$ | 17 | 7 | 18 | 4 | 17 | 3 | NaN | NaN | 19 | 2 |
| | $t_{mode}$ | 10 | 2 | 10 | 1 | 6 | 1 | NaN | NaN | 12 | 1 |
| 500 | $\Gamma$ | NaN | NaN | NaN | NaN | 23 | 2 | NaN | NaN | 29 | 1 |
| | D | NaN | NaN | NaN | NaN | 16 | 3 | NaN | NaN | 17 | 2 |
| | $t_{mode}$ | NaN | NaN | NaN | NaN | 7 | 1 | NaN | NaN | 12 | 1 |
| 1000 | $\Gamma$ | NaN | NaN | NaN | NaN | 21 | 3 | NaN | NaN | 36 | 2 |
| | D | NaN | NaN | NaN | NaN | 21 | 5 | NaN | NaN | 28 | 4 |
| | $t_{mode}$ | NaN | NaN | NaN | NaN | 3 | 1 | NaN | NaN | 9 | 1 |

**Table C2.** Amundsen Basin

| Depth (m) | TTD parameter (years) | 2011 218 val | unc | 2012 – val | unc | 2015 81 val | unc | 2020 – val | unc | 2021 16 val | unc | 2021 20 val | unc |
|---|---|---|---|---|---|---|---|---|---|---|---|---|---|
| 300 | $\Gamma$ | 25 | 4 | NaN | NaN | 21 | 3 | NaN | NaN | 26 | 1 | NaN | NaN |
| | $\Delta$ | 22 | 8 | NaN | NaN | 21 | 6 | NaN | NaN | 15 | 2 | NaN | NaN |
| | $t_{mode}$ | 5 | 1 | NaN | NaN | 3 | 1 | NaN | NaN | 11 | 1 | NaN | NaN |
| 500 | $\Gamma$ | NaN | NaN | NaN | NaN | 23 | 3 | NaN | NaN | 27 | 2 | 26 | 2 |
| | D | NaN | NaN | NaN | NaN | 22 | 6 | NaN | NaN | 21 | 3 | 23 | 4 |
| | $t_{mode}$ | NaN | NaN | NaN | NaN | 4 | 1 | NaN | NaN | 7 | 1 | 5 | 1 |
| 1000 | $\Gamma$ | NaN | NaN | NaN | NaN | 11 | 3 | NaN | NaN | 32 | 2 | 30 | 2 |
| | D | NaN | NaN | NaN | NaN | 15 | 7 | NaN | NaN | 27 | 4 | 21 | 3 |
| | $t_{mode}$ | NaN | NaN | NaN | NaN | 1 | 1 | NaN | NaN | 7 | 1 | 9 | 1 |

**Table C3.** Makarov Basin

| Depth (m) | TTD parameter (years) | 2011 226 val | unc | 2012 – val | unc | 2015 101 val | unc | 2020 – val | unc | 2021 28 val | unc |
|---|---|---|---|---|---|---|---|---|---|---|---|
| 300 | $\Gamma$ | 23 | 3 | NaN | NaN | 22 | 1 | NaN | NaN | 30 | 1 |
| | $\Delta$ | 9 | 4 | NaN | NaN | 11 | 3 | NaN | NaN | 16 | 2 |
| | $t_{mode}$ | 15 | 2 | NaN | NaN | 11 | 2 | NaN | NaN | 14 | 1 |
| 500 | $\Gamma$ | NaN | NaN | NaN | NaN | 26 | 2 | NaN | NaN | 31 | 1 |
| | D | NaN | NaN | NaN | NaN | 13 | 3 | NaN | NaN | 15 | 2 |
| | $t_{mode}$ | NaN | NaN | NaN | NaN | 13 | 2 | NaN | NaN | 16 | 1 |
| 1000 | $\Gamma$ | NaN | NaN | NaN | NaN | 47 | 2 | NaN | NaN | 42 | 1 |
| | D | NaN | NaN | NaN | NaN | 26 | 2 | NaN | NaN | 21 | 2 |
| | $t_{mode}$ | NaN | NaN | NaN | NaN | 21 | 1 | NaN | NaN | 21 | 1 |

**Table C4.** Canada Basin

| Depth (m) | TTD parameter (years) | 2011 | | 2012 | | 2015 | | 2020 | | 2021 | |
| | Station | – | | – | | 56 | | CB5 | | – | |
| | | val | unc | val | unc | val | unc | val | unc | val | unc |
| --- | --- | --- | --- | --- | --- | --- | --- | --- | --- | --- | --- |
| | $\Gamma$ | NaN | NaN | NaN | NaN | 35 | 1 | 31 | 2 | NaN | NaN |
| 500 | D | NaN | NaN | NaN | NaN | 9 | 1 | 5 | 2 | NaN | NaN |
| | $t_{mode}$ | NaN | NaN | NaN | NaN | 29 | 0 | 29 | 0 | NaN | NaN |

*Author contributions.* AMW performed the conceptualization, investigation, data curation, formal analysis, visualization, and wrote the original draft. AP and MS supported the conceptualization and data curation and performed writing (review and editing). CV and MC supported the methodology and formal analysis and performed writing (review and editing). TT acquired funding, provided resources and performed writing (review and editing). NC supported the conceptualization, and investigation, acquired funding, and performed writing (review and editing). AMS measurements were performed by CV, MC, and NC.

*Competing interests.* The authors declare that they have no conflict of interest.

*Acknowledgements.* We would like to thank all people involved in the sampling activities across the Arctic Ocean and the subsequent sample processing and measurements, providing the available $^{129}$I and $^{236}$U data used in this study. Special thanks to the captain, crew, and all scientists of the SAS2021 expedition, and particularly to Lennart Gerke and Yannis Arck for collecting seawater samples for $^{129}$I and $^{236}$U. We acknowledge the support of the DFG (Deutsche Forschungsgesellschaft), financing the project "Der arktische Ozean 2020 — Ventilationszeitskalen, anthropogener Kohlenstoff und Variabilität in einer sich verändernden Umgebung" (TA 317/8-1 and AE 93/21-1) and also providing the opportunity to participate in the SAS-Oden 2021 cruise. We also acknowledge ARICE (Arctic Research Icebreaker Consortium, grant number 730965), the Hasselblad Foundation (Contract No. 2019-1218) and the Swedish Polar Research Secretariat (SPRS) for their support during the 2021 cruise. Thanks to Kayley Kündig for assistance in sample processing. Large parts of this work were carried out under the TITANICA project (PI N. Casacuberta), which is funded by the European Research Council (ERC) under the European Union's Horizon 2020 research and innovation programme (Grant 101001451). N. Casacuberta is also funded by the Swiss National Science Foundation (PR00P2-193091-TRACEATLANTIC). A.-M. Wefing acknowledges funding from the SNSF Postdoc.Mobility fellowship TRACPAC (Project number P500PN_217968). In this study, color maps from Thyng et al. (2016) and Crameri (2023) were used for the visualization of results. We also thank two anonymous reviewers for their comments which improved the manuscript.

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
