# Peer review of "Changes in water mass composition and circulation in the central Arctic Ocean between 2011 and 2021 inferred from tracer observations"

_EGUsphere, 2025_

## Author Comment (AC1)

Manuscript: Circulation timescales of Atlantic Water in the Arctic Ocean determined from anthropogenic radionuclides

**Response to RC1**
**(comment posted 04 June 2025, commented pdf of the manuscript)**

Dear RC1,

Thank you for your review and helpful suggestions.

This response letter includes responses to the comments in the pdf and the corresponding changes made to the manuscript. We would like to point out that following the suggestions by RC2, the manuscript has been substantially restructured and large parts have been moved from the discussion to the introduction, methods, or results part. Furthermore, figures and corresponding text have been changed to refer to practical salinity instead of absolute salinity throughout the manuscript.

On behalf of all co-authors,

Anne-Marie Wefing

*Note: Reviewer comments in black (lines refer to the original manuscript), response in blue*

This study represents an important contribution to our understanding of Atlantic Water distribution and circulation in a region marked by the complex confluence of multiple water masses. The scientific approach is robust, and the organization of material and quality of figures are both excellent. Most of my comments are stylistic in nature, with a few minor typographical corrections.

Introduction: several references suggested

We thank the reviewer for suggesting many helpful references. The references in the introduction have been revised and several have been added. Note that the introduction has been partly rewritten following suggestions from RC2.

Line 100: suggested to add the Lincoln Sea in Figure 1. Has been added.

Line 221: typo has been corrected.

Line 224: "Considering" has been substituted by "At"

Line 228-229: has been changed to "Above 100 m, the decline in I-129 and U-236 in Section 2 (left half of the section) was clearly evident and coincided with a decrease in

practical salinity from around 32.5 g kg⁻¹ to around 30.5 g kg⁻¹ at 50 m depth." as suggested.

Lines 235-237: temperature values and differences have been included and the sentence has been changed to: "PSW from Section 2 had slightly higher temperatures (difference around 0.5∘C) compared to Section 1, for salinities above 32 (see also Section 4.2). In AAW and below, the opposite was observed, with Section 1 stations showing higher temperatures (difference up to 1∘C) compared to Section 2 (Fig. 5b). Highest AAW temperatures of about 1.5∘C (conservative temperature) were observed at station 5 in the Nansen Basin, closest to the Atlantic Water inflow region."

Lines 244-245: stylistic suggestions have been incorporated.

Line 253: concentrations range has been added and stylistic suggestions incorporated.

Line 263: the range of differences has been stated.

Line 266: sentence has been changed to "Along Section 2, [...]" as suggested.

Line 279: To me "Atlantic Water Circulation" feels little narrow given the scope of the subsections. Suggesting changing 4.1 tittle to something more broad in scope (Arctic Ocean Water Mass Circulation in 2021, Water Mass Provenance? Pathways, and Ventilation Timescales in 202

Following the suggestions by RC2, the results and discussion sections have been restructured and titles have been changed accordingly.

Line 301: sentence has been removed and the discussion about the Nansen Basin has been integrated with section 4.1

Lines 304-305: see comment above. The publication year has been added to Perez-Tribouillier et al. (2025).

Line 313: sentence has been removed and the global fallout tracer signal has been included in section 2.2 instead.

Line 315: has been changed to "... meteoric water (net precipitation and river runoff) ..." as suggested.

Line 317-320: stylistic suggestions have been incorporated, suggested references have been added.

Line 321: consider adding depth level (e.g. highest in the upper x meters) and spatial distribution

Has been added, the sentence now reads: "Meteoric water fractions were up to 0.12, highest in the upper 50m in the Amundsen and Makarov Basins."

Line 326: "Consider being consistent in the use of either fractions or percentages when discussing water mass composition. I realize this is a pedantic point—apologies"

Thank you for pointing that out, we now report the results of fractions instead of percentages.

Line 326: "consider adding a note on brine rejection (negative SIM)"

Has been added in line 417.

Lines 329-331: sentence has been changed to "Pacific Water fractions have been estimated using various parameters (e.g., silicate, nutrient ratios, gallium), each with its own strengths and limitations. These methods remain debated and carry large uncertainties (e.g., Alkire et al., 2015, 2019; Whitmore et al., 2020), however, they still serve for an overview of the spatial distribution of Pacific Waters across the sampling area." as suggested.

Line 332: values have been added as suggested.

Line 337: uncertainty levels have been added.

Lines 352-355: This has been rephrased and moved to section 3.3.

Lines 357-359: sentence has been changed to "... which in the Eurasian Basin is formed through repeated winter convection cycles north of the Barents Sea, as well as by advection of low-salinity shelf water, primarily from the Laptev Sea ..." as suggested.

Line 362: reference Anderson et al. 1994 has been added

Line 369: sentence has been changed to "Based on I129 and U236, we conclude that..."

Line 372: has been changed to "large variability" as suggested.

Lines 376-377: this has been rewritten and mentioned in the methods already.

Line 381: might be helpful to specify this refers to the boundary current flow ie along the continental slope.

This has been specified (line 544).

Lines 389-390: sentence has been changed to "The TTD parameters $\Gamma$ and $\Delta$, the $\Delta/\Gamma$ ratio, and the mode age tmode derived for SAS21 and JOIS 2020 (Canada Basin) are presented in Fig. 7." (line 340-341).

Line 421: stylistic suggestions have been incorporated (line 350).

Lines 430-432: sentence has been simplified to: "Samples from station 46 (dark red) plotted close to or directly on the mixing lines." (line 354-355).

Line 434: has been changed to "(Eq. (4))" according to journal convention.

Line 436: consider moving this sentence up to where you introduce the bimodal TTD.

Sentence has been moved to subsection 2.5.

Lines 439-446: this part has been rewritten and partly moved to subsection 3.4.

Lines 454-456: sentence has been simplified to "The I129 distribution in surface waters (approximately 10-30 m, depending on sample availability) was characterized by higher concentrations in the Amundsen Basin compared to the Makarov Basin and north of Greenland (Fig. 9a)." as suggested.

Line 464: stylistic suggestion has been incorporated.

Line 495: "ongoing shift" might suggest that the AW/PW front continued advancing into the EB but the following sentence clarifis that it remained stable. Maybe: the front has not regressed back towards the Mendeleev Rise..

Sentence has been changed to "Our analysis of I129 concentrations between 2011 and 2021 indicated that the position of the front still changed between 2011 and 2015, but remained stable between 2015 and 2021".

Line 498: maybe TPD should be briefly described here on in the intro

The TPD is now introduced in section 1.2.

Lines 530-531: parts of this section have been moved to the introduction and rewritten.

Line 546: has been changed to study region.

Line 563: "lateral" has been added as suggested.

Lines 732-733: check out https://doi.org/10.1175/JPO-D-20-0190.1

Thank you for the suggestion. The reference to Morison et al. 2021 was included in section 1.1 and 1.3, about the relation between a strong boundary current and a positive AO index.

---

## Author Comment (AC2)

Manuscript: Circulation timescales of Atlantic Water in the Arctic Ocean determined from anthropogenic radionuclides

**Response to RC2**
**(comment posted 10 June 2025)**

Dear RC2,

Thank you for your review and helpful suggestions.

This response letter includes responses to your comments, outlined below. The manuscript has been substantially restructured and large parts have been moved from the discussion to the introduction, methods, or results part. Furthermore, figures and corresponding text have been changed to refer to practical salinity instead of absolute salinity throughout the manuscript. We hope that the restructured and rewritten manuscript now transfers the main messages better.

On behalf of all co-authors,

Anne-Marie Wefing

*Note: Reviewer comments in black (lines refer to the original manuscript), response in blue*

The manuscript by Wefing et al. presents interesting results obtained from radionuclides that allow to further study changes in the Arctic Ocean circulation. I believe that the manuscript will be an important contribution to Arctic oceanography after major revisions. While I have few comments on the methods, results, and conclusion, I believe that the presentation of this study, i.e., the writing and organization of the manuscript has to be substantially improved. I am not referring to the English language but to the structure of the manuscript and the introduction of key concepts. In the following I will give examples of improvement. However, I will refrain from too many detailed word-by-word comments as I expect the text to be largely adapted after revisions. It is my strong believe that a large reconstruction of the manuscript will improve the flow of the text drastically and allow the great results to be better grasped by the reader.

Major points:

- Large parts of the discussion are actually results of methods. Here is a list that is likely not complete:

- o Section 4.1.1: First paragraph belongs into methods; second paragraph is results

- o Section 4.1.3: First paragraph is a mixture of methods, results, and figure legend, the beginning of the second paragraph is also rather results.

- o Section 4.1.4: Second paragraph is methods; second paragraph is results; last paragraph is also full of results

- o Section 4.2.1: First paragraph contains many results; second paragraph is a mixture of results and methods; third paragraph also contains many results

- o Section 4.2.2: First paragraph is methods; second paragraph is partly results; third and fourth paragraphs are results.

- o Section 4.2.3: Large parts of the first paragraph and the second paragraph should probably be introduced in the Introduction

As a consequence of having large parts of the results, methods, and introduction in the discussion section, parts of the conclusion repeat or summarize the results. After all sections are clearly separated, the Discussion will likely substantially decrease and now repetition will be needed in the Conclusion and also lead to a substantial reduction of its length.

The manuscript has been restructured and large parts of the discussion have been moved to the results section, some to the methods or introduction sections. The conclusions have been shortened. Figures, incl. those in the appendix, have been rearranged to fit to the new structure.

The Introduction does not seem to fit well enough to the rest of the paper. The scope of the study appears to me the placement or change in the shift of the Atlantic-Pacific waterfront in the Arctic Ocean. Reading the Introduction it remains unclear why this position actually matters. I also found that couple of concepts are not well enough introduced. The first paragraph is about atlantification, the second one about the flow of Atlantic and Pacific waters in the Arctic Ocean, the third one about the depth structure (difficult to follow, please see detailed comments in the minor comments part), the last paragraph is somehow about the connection to the global ocean and turns into a description of radionuclide studies in the Arctic (these two topics do not fit well into one paragraph). Afterwards there is a new section in the Introduction going into details of radionuclides, starting with a paragraph about the sources of two such radionuclides, the second one is about the flow of these radionuclides into the Arctic, the next paragraph is about classical tracers such as CFC-12 and SF6, the next one is about the difference between the classical to non-classical tracers, the last one is about the aims

of the study. While I think that the second subsection is well structured, I am left confused with the first one. After reading it, I am missing information about the importance of the study here. What is the problem that is going to be solved? What is the question? Afterwards, in the discussion, that is introduced when discussing literature about shifts in the frontal zone between Pacific and Atlantic water. I believe it would be crucial to bring all this up here. It would also help for the reader to explain why it matters if the boundary between Atlantic and Pacific waters moves a bit. At the moment, it is not even clear to me but I still find the application of the new tracers interesting. However, despite being an interesting application, the Introduction leaves me in the unknown why I should care about this application apart from a demonstration of the methods. It would therefore be of great importance to better explain the importance of this question. To do that, I would probably structure the first part as follows: 1) The Arctic is changing fast: changing sea ice, changing primary production, changing acidification, changing temperature, changing carbon sink, etc. 2) A large amount of these changes is driven by changes in the circulation (try to bring a few examples), Finish with a sentence stating that knowledge of the circulation and its changes is limited due to limited numbers of observations, 3) describe what we know about the flow paths, 4) describe what we know about the vertical column, 5) describe known changes over the last decade or two and known uncertainties. Say that it is a problem not to know because it makes it hard to assess changes and to make projections (if that is the main reason why we need these tracers). You can finish than with a brief sentence that tracers, such as radionuclides and CFCs can help. Then you can move to the second part of the Introduction. To me, that sounds more intuitive, but maybe you have a better idea or prefer your text.

The introduction has been restructured. The focus of this study is not particularly the AW-PW front, but the overall changes in the distribution and circulation times of Atlantic Water over time, and how anthropogenic radionuclides can be used to this purpose. This hopefully became more clear also through the restructured discussion. The detailed description on the use of I129 and U236 as tracers in the Arctic has been moved to the methods part.

Usually, I would not add this to the major comments but the word 'note' is used far too many times in my opinion. I am not sure if it is correct, but I had learned a while ago during my studies that 'note that' or 'it should be noted that' should not be used in scientific writing. If something is noteworthy, I was told, it should be written in a way that the reader knows it is noteworthy. Given the numerous times this word is used, I believe removing it can substantially reduce the text and improve readability.

Sentences containing the word 'note' have been rewritten, please refer to the track changes manuscript for details.

Minor comments:

- The first sentence of the abstract is hard to understand as it remains very unclear what and how circulation changes contribute to Atlantification and what processes of Atlantification are meant. It is also unclear what role the Arctic circulation plays in these processes.

- The second sentence of the abstract talks about recent variability and trends. It is unclear what variability and trends the authors are referring to. It is also unclear why the circulation needs to be better understood

- Sentence three in the abstract could be improved by relying on the 'old-before-new' principle. If the sentence was started with: "Here, we investigate the Atlantic water ciruculation times and mixing in the Arctic Ocean to better understand the mixing between Atlantic and Pacific Waters using…", the reader would first be referred to what they know from the previous sentence before going to the new subject, i.e., radionuclides and TTD.

- The sentence from line 8 to 9 does not fit here. First you write about the data from 2021, then you talk about changes in this sentence, then you go back to 2021 and finally you go back in line 15 to temporal changes. I'd suggest moving the sentence in line 8 to 9 to line 15. That would also allow you to safe some words in line 15.

- Line 11: I am not sure what below refers to? Below the surface? Please precise.

- Line 11: I am also not sure what similar refers to, similar to what? And comparably high compared to what?

- Line 14: Similar: higher circulation times than what?

- Line 15 to 16: Please indicate what kind of shift you find, a shift towards the Eurasian basins or another direction. Just a shift, is not informative enough. Also please indicate the distance of the shift.

- Last sentence of the abstract: I would re-order it and say that the increase in mode ages suggests a slowdown of the AOBC, which is in line with recent studies basied on gas tracers.

The abstract has been rewritten, taking the comments above into account. Details have been clarified. Since we do not calculate absolute water mass fractions from I129, but only use the concentrations to derive qualitative changes, we refrain from giving a quantitative estimate of the shift of the AW-PW front.

- Section 1.1: First paragraph: There are often missing links between the sentences. The first one is about Arctic amplification. The second one is direction about a special reason and a new concept, Atlantification. It remains unclear how the third sentence connects to the second one. The fourth sentence then connects to the second one although the connection only becomes clear at the end of the sentence. The last one then connects to the one above. Please try to restructure the paragraph to better connect the sentences and think what the topic of this paragraph really is, Arctic amplification as mentioned in the first sentence or Atlantification as this is the topic of most sentences in the paragraph.

  As mentioned above (under major comments), the introduction has been restructured and rewritten.

- Line 37: How is the upper layer defined? The surface layer? But Pacific winter waters also sink below? Please be precise:

  This has been rewritten as follows: "The PSW layer is highly stratified and generally entails Pacific Water, river runoff, sea-ice meltwater, as well as transformed Atlantic Water. Pacific Water enters the Arctic Ocean through Bering Strait, mainly resides in the Canada and Makarov basins, and is restricted to the PSW layer due to its low density…"

- Paragraph 3 of the Introduction: It is hard to follow this paragraph, although I am familiar with the physical oceanography in the Arctic. I have no precise advise how to restructure it at hand right now, but I'd strongly suggest to give it another try. In line 43, it is for example not clear what 'it' refers to, in line 44 it would be helpful to be more quantitative and not just write 'deeper' (is it deeper than something else?). I am really puzzled after this paragraph.

  This paragraph about the Arctic Ocean water column has been rewritten and slightly extended to include the mid-depth Atlantic layer as well as intermediate and deep waters.

- Paragraph 4: The topic sentence is about connections to the global circulation and the paragraph about radionuclides. It remains unclear how the connection works, and how changes impact regions beyond the Arctic Ocean. Please expand this explanation substantially and then cut the paragraph once you go to the past tracer studies.

  The description about the connection of the Arctic Ocean to the global circulation has been moved to the first paragraph of the introduction and extended.

- Section 1.2, first paragraph: I am confused that U236 is mainly introduced via global fallout but still the liquid release is dominant in the Arctic. Could you clarify this please?

  This has been clarified in the text (line 141-144 of the revised manuscript).

- Line 79 and 80: You say that the concentration and distribution was assessed in the Canada Basin but you do not say what was found and why this matters. Without that information, this sentence is not really helpful to the reader.

  This sentence has been removed.

- In the major point I have not really said anything about paragraph 3 of section 1.2, but I would likely introduce these tracers first in the section as they are older and then you can introduce your new tracers with their distinct advantages. This would likely even increase the importance of the new tracers.

  This section (now section 1.3) has been restructured.

- I'd suggest cutting the sentence from line 85 to 87 in two sentences for improved readability.

  The sentence has been split in two.

- Line 116: Do you really need the second reference? If it is also described in the first one, I am not sure what the second reference adds. If it adds something, please describe its additional value. Same goes for line 124.

  The second reference has been removed in both places.

- Lines 134 and 135: The part of the sentence that starts with ", which are listed ... " can just be replaced by "(Table 1)" to safe words.

  This has been shortened.

- Lines 194 to 196 can easily be added to the figure legend and to not need to be part of the main text.

  This part has been moved to the figure legend of Figure 3.

- Figure 4: I found it very hard to distinguish the colors of this colormap. It might be an issue with my eyesight but I'd like to encourage the authors to adjust or change the colormap.

  All colormaps were chosen according to recent scientific studies (Thyng et al., 2016, https://doi.org/10.5670/oceanog.2016.66; Crameri et al., 2020, https://doi.org/10.1038/s41467-020-19160-7). We are not sure which panel/colormap of Fid. 4 the reviewer is referring to. We have increased the size of the datapoints in panel c and d.

- Line 198: It might be more easily to read if 'Section 1' would be replaced by a more descriptive name like 'Eurasian section' or 'Section through the Eurasian basins'.

  This has been rewritten to "For all profiles from the Eurasian Basin section (Section 1), …" (line 235).

- Line 211: In the Introduction, these fallout levels were never quantified.

  The global fallout levels for both radionuclides have been added to section 2.2 (line 146).

- Line 214: Similar to above, I'd suggest replacing 'Section 2' by a more descriptive name.

  This has been rewritten to "For the section through the Makarov Basin and north of Greenland (Section 2), …" (line 251).

- Line 221: Is the word significant used here to describe statistically significant values? If not, I'd suggest replacing it by substantial or a similar word to avoid misunderstandings.

  Has been replaced by "substantially".

- Lines 225 to 226: Please try to be more quantitative, maybe in %?

  Decrease of I129 concentrations in % has been added.

- Line 229: Is this also related to a change in the MLD?

  The change in salinity is related to a change in water mass composition, i.e., more Pacific Water in the Makarov Basin. The same holds for the change in I129 and U236, which are diluted by Pacific Water and freshwater in the MB. The MLD decreases from EB to MB, the water column is more stratified in the western Arctic (MB). We do not assume that the change in MLD generally leads to a change in tracer concentrations.

- Line 236: especially above 32 or only at salinities above 32?

  This applied only to salinities above 32, so "especially" has been removed.

- Line 241: This is not really a topic sentence. It would be easier to read to state here the main message of the paragraph and not just the figure legend of the figure that is going to be described.

  The first sentence has been rewritten to "Different sampling regions showed different relations between I129 and salinity (Fig. 5c)." (line 280).

- Lines 264 and 265: Instead of writing that differences were observed, you could just directly say what differences where observed to safe space and to safe time of the reader.

  The decrease of I129 concentrations has been stated in the first sentence (line 303-304).

- Line 315: It might be helpful to explain what meteoric means. Please do not mind that comment if I am just not well educated and should understand it.

  This has been further clarified: "... meteoric water (net precipitation and river runoff)..." (line 397).

- Line 330: Some N:P ratio is introduced and difficulties are mentioned. However, this method is never mentioned or explained before. Please explain it in the Introduction or the Methods. Otherwise, it is very hard to understand.

  The N:P ratio as a method to determine water mass fractions is mentioned in line 413 (this part has been rewritten following suggestions of RC1). This is an established method which has been used in numerous studies, several of which are listed as references. Here, it is only used as a comparison to provide context for the I129 concentrations. Since our study does not focus on the use of new tracers to quantify water mass fractions, we decided not to introduce this method in the introduction or methods and suggest not to put too much focus on it. We now use the term "N:P method" consistently when referring to this method.

- Line 337: Please excuse me if I misunderstood the method, but could the difference in the I129 concentrations not simply indicate a difference in the age of the waters but the waters could still have the same mix of Pacific and Atlantic waters, just older Atlantic waters?

  Older Atlantic Waters indeed have lower I129 concentrations. However, it is very unlikely that this explains the difference in I129 concentrations observed between the Amundsen and Makarov Basin due to the following:

  - Older waters would have higher U236 concentrations, which are not observed.
  - Surface waters in the Makarov Basin would have to be about 15-20 years older than in the Amundsen Basin to explain the observed decrease of about 50% in concentrations (assuming an age of about 15 years in the Amundsen Basin: 50% of the I129 concentrations found in the surface layer input function 15 years prior to sampling in 2021, i.e., 2006, would correspond to input function concentrations found around 1990). This is not observed in other studies.

- o Waters in the halocline layer have similar I129 concentrations in both basins. Hence, waters in the upper 50m would have to be substantially older than at around 100m in the Makarov Basin to explain the lower I129 concentrations, while this would not be the case in the Amundsen Basin. Furthermore, there is no evidence for a substantially faster circulation of halocline waters compared to the waters at the surface.

  We added a sentence to point this out (line 423-425).

- It would also not harm to have a bit of caveats on the number of sampled stations, i.e., how robust is the determination of the shift in the position of the fronts.

  This has been added to the end of section 4.2.

- Line 371: Think about properly introducing the NO parameter in the Methods

  As for the N:P ratios, we consider the NO parameter more as an auxiliary tracer to the radionuclide dataset and therefore decided not to introduce it in the introduction or methods. However, the comparison to the NO parameter has been slightly extended (last paragraph in subsection 4.4).

- In some places you use absolute salinities and in other places practical salinity units. Please try to use one of them consistently everywhere, preferably practical salinity units (personal taste).

  We now use practical salinity consistently throughout the paper. Figures and text have been changed accordingly.

- Lines 396 and 398: It would be helpful to introduce the concept of mode and mean ages and the associated differences earlier.

  We are not sure what the reviewer is referring to. Maybe this is referring to lines 403-404 in the submitted manuscript ("Previous studies suggested the mode age as a more suitable age measure for the lateral transport of Atlantic Water in the Arctic Ocean compared to the mean age (Smith et al., 2011; Wefing et al., 2021)."). This sentence has been moved to the methods (subsection 2.5).

- Lines 418 and 419: The last sentence in this paragraph does not help a lot. What does this refer to? And please mention the implications as it is not informative without that information.

  This sentence has been removed and the reference has been included in the sentence before.

- Lines 455 and 456: Higher than what? Lower than what?

  This sentence has been rewritten (lines 457-459).

- Lines 487 to 489: I think this discussion merits a bit more detail. I do not understand immediately how these different atmospheric patterns affect the Arctic Ocean circulation.

  This has been moved to the introduction (section 1.1) and expanded.

- Line 546: This is where the discussion of your results really starts while the text before is mainly an Introduction.

  Section 4 (Discussion) has been rearranged and large parts have been moved to methods/results as suggested.

- Line 558: Maybe add also higher spatial coverage here.

  Has been added.

- Line 559: Do you mean data here (including also models) or do you explicitly refer to observations. If it is the second, please use observations.

  Has been changed to "observations".

- Line 568: I think it would be better using the active voice here.

  Has been changed to "We attributed this to …".

- Line 573: At the end of this line, it looks as if a new paragraph is starting.

  Has been split in two paragraphs.

Again, I want to re-iterate that I find this paper very valuable for the understanding of the Arctic Ocean circulation. It is especially because of this value that I believe that a large investment in the writing and structuring could be very valuable and really help to transfer the main messages of this paper.

We agree with the reviewer and hope that new structure of the paper improved its readability and conveys the main messages more clearly.

---

## Referee Report (RR1)

General Assessment

This is a well-executed and ambitious study. The methodology is robust and clearly presented, and the figures—both in terms of quality and content—are excellent. However, the manuscript's organization and presentation could be improved. Given the amount of data and analysis presented, these adjustments are non-trivial but will significantly enhance readability. That said, some of these suggestions are inherently subjective.

**Title.** Recommendation: The current title focuses narrowly on Atlantic Waters, but the manuscript covers a broader range of water masses. Consider adjusting the title to reflect this broader scope, even in the abstract, where Atlantic water is only discussed in the final paragraph. Options might include:

**Section 3**.

Terminology: Use terms consistently—e.g., choose either potential temperature or temperature, and salinity or practical salinity throughout.

Quantification: Reduce the use of vague comparative language such as "around," "slightly more," or "much higher." Instead, refer directly to measured values, as these are available in the figures.

Narrative Style: The figures are beautiful and highly informative—consider letting them speak more for themselves. The current text is thorough, but some passages feel overly explanatory, which detracts from readability.

Section 3.3–3.6: These could potentially be moved to the Discussion, as they begin to interpret rather than purely describe.

Idea: It would be helpful to define Sections 1 and 2 by station numbers and geographic location early in the manuscript. This would eliminate the need to repeatedly remind the reader later. Maybe abbreviate to S1 and S2..

**Section 4.**

Some subsections (4.1, 4.5, 4.6) feel underdeveloped. Suggest adding a short note on uncertainty due to data resolution.

Consider explicitly addressing the uncertainty introduced by the spatial and temporal resolution of the datasets. This would strengthen the interpretations, particularly those related to circulation patterns.

**Recommendation: Recommended for publication.**

**Specific Suggestions and Comments**

**Lines 1–20**: The abstract would benefit from streamlining and reorganization. Consider introducing both new and historical datasets earlier, as both are essential for analyzing temporal changes — and should only be introduced once. For example, move the sentence starting with "Temporal changes in the circulation [...] between 2011 and 2021" (line 15) closer to the beginning to clarify the study's focus.

**Line 5**: Remove "mainly" — both datasets are necessary to characterize Atlantic Water (AW) changes.

**Line 28**: Add original citations for the stated information.

**Line 32**: Replace "driven" with "linked to" or "associated with" — warming is likely the true driver.

**Line 41**: Replace "was" with "is," or clarify the time frame. Consider citing Smith et al. (2021) here, as it discusses AO-related boundary current changes.

**Line 47**: Use "the two *primary* large-scale circulation patterns."

**Line 49**: Add original references.

**Lines 70–73**: Remove "extensive." Add original sources: e.g., Polyakov et al. (2005) use T anomaly propagation, Dmitrenko et al. (2008), Woodgate et al. (2001), Li et al. (2021). For tracers, include Frank et al. (1998); Broecker & Peng (1982) may not be the best reference for CFCs/$SF_6$.

**Figure 1**: Consider color-coding historical data by cruise or sampling year for clarity.

**Line 89**: Also cite Smith et al. (2011).

**Line 95**: Add a supporting citation.

**Line 98**: Add a reference for the statement about timescales "on the order of decades or below."

**Line 103**: Cite a source for the convergence of waters before exiting via Nares or Fram Strait.

**Line 107**: Replace "Arctic system" with a more specific term (e.g., Arctic Ocean circulation or stratification).

**Line 139**: "Data from *six* expeditions."

**Lines 142–145**: Use original references.

**Lines 158–159**: Remove the phrase referencing Raimondi et al. (2024) and Payne et al. (2024).

**Line 180**: Rephrase to "This model calculates mixing lines…" for clarity.

**Line 195**: Clarify whether "mid-depth" here matches the typical depth range used for AW elsewhere in the manuscript.

**Lines 245–249**: Consider revising to: "Station 5 shows a sharp decrease between 700 and 1000 m."

**Line 280**: "Color-coded" may be redundant — consider removing.

**Line 306**: "Close to the temperature maximum" — is this referring to the AW core? Clarify.

**Line 319**: Sentence beginning with "The compilation of available 129I data…" may not be necessary — consider cutting.

**Line 330**: Rephrase: "To account for differences in sampling depths and water mass composition…"

**Line 333**: "Very low" may be unnecessary — consider deleting.

**Lines 343–345**: Replace "shallow" with "surface waters" for consistency.

**Line 365**: [Appears incomplete — please confirm if something's missing.]

**Line 411**: Clarify which layer is being referred to — "surface" vs. AW could be confusing. Add original Rudels reference.

**Line 431**: Rephrase and support with citations: e.g., "Because 129I is introduced via Atlantic waters and diluted by Pacific inflow, its distribution across the SAS2021 section reflects the Atlantic–Pacific water mass composition in PSW."

**Line 531**: Sentence on TTD-derived mode ages not needed in the Discussion — already addressed in Results.

**Lines 564–571**: Quantify terms like "slowdown" and "increasing ages" for accessibility to broader readership.

**Lines 598–600**: Consider omitting this restatement if already covered earlier.

**Line 615**: This is the first 129I and 236U dataset north of Greenland — highlight this earlier, perhaps in the abstract.

**Line 651**: Add a sentence defining equation terms: e.g., "…where fA, fPac, fSIM, and fMet represent the fractions of Atlantic Water, Pacific Water, sea-ice melt, and meteoric water, respectively."

---

## Author Response (AR2)

Manuscript: Circulation timescales of Atlantic Water in the Arctic Ocean determined from anthropogenic radionuclides

**Response to editor comment on revised version of the manuscript:**

Dear Editor,

Thank you for your comments and suggestions.

We have incorporated your comments into the revised version. Please see point-by-point responses below.

On behalf of all co-authors,

Anne-Marie Wefing

Both reviewers were overall positive about the submitted manuscript (many thanks to both for their constructive comments), with reviewer 1 suggesting only minor revisions, but reviewer 2 suggested a major revision including updating the structure to improve the clarity of the manuscript. I thank the authors for engaging well with the reviews. However, given the major changes, I would like to ask the reviewer(s) to take another look at the revised manuscript. But before that, I have a few minor issues that I would like to ask the authors to resolve. For the tracked-changes version, please could you continue working from the current tracked-changes document, so that all changes since the original submission are shown.

The tracked-changes document contains all changes made since the original submission of the manuscript.

The abstract should essentially follow the structure of the paper in a shortened format – the revised version lacks any introduction / context / motivation at the start, instead starting directly with the methods. Please update to add a couple of introductory sentences.

We added the following two sentences in the beginning of the abstract:

"The Arctic Ocean is changing rapidly and Atlantic Water circulation plays a key role in the warming, sea-ice decline, and ecosystem changes observed in the Arctic. Still, we only have limited understanding of the pathways and circulation times of Atlantic Water in the Arctic Ocean and how they evolve over time."

Reviewer 2 was concerned about the lack of a clear motivation or research question. Within the introduction, the revised section 1.1 is where this is addressed. In my view, there is still not a clear research question here. The section mentions a few relevant ideas, but the reader still has to join the dots and guess exactly what the question might

be. I suggest the authors try to make the question and motivation more explicit here (or elsewhere within the introduction), and to more clearly connect these points together, before I send this back out to review.

We changed the last paragraph of section 1.3 to the following to clarify the research objectives of this study:

"This study aims to assess the circulation pathways and timescales of Atlantic Water in the central Arctic Ocean in 2021, with particular focus on the Lincoln Sea north of Greenland - a strategic location where waters from the Eurasian and Amerasian Basin converge before exiting the Arctic through the Nares or Fram Strait. By combining new I129 and U236U data collected in 2021 with historical data from similar locations, we constrain transport times and mixing processes of Atlantic-derived waters, characterize the composition of surface waters with particular emphasis on the extent of Pacific Water, and evaluate temporal changes in circulation over the decade from 2011 to 2021. This study contributes to the understanding of how changes in the Atlantic Waters entering the Arctic Ocean affect the Arctic system and how these waters mix with Pacific-origin waters in the upper water column."

Another major concern of reviewer 2 was that many parts that belonged in the methods or results sections were instead in the discussion. It looks like many of these issues have been fixed in the revised manuscript. However, could the authors please double check on section 4.3 – all except the final paragraph might be more results than discussion.

We have moved parts of section 4.3 to the results, to a new section with the title "Temporal Changes in 129I Concentration in PSW" (section 3.3). We have split former Fig. 10 into a new Fig. 6 (in section 3.3) and a new Fig. 11 (in section 4.3).

Please could the authors check carefully how dates are referred to throughout, to make sure they are clear. For example, in lines 72-74, does 'over the past 16 years' refer to 2009-2025, or rather to 2005-2021 (as the study period was 1991-2021)? In line 561, was the decrease in the strength of the boundary current observed from the mid-1990s to 2015 or were the tracer studies written from the mid-1990s to 2015?

Line 74: "over the past 16 years" changed to "between 2005 and 2021"

Line 561: changed to "A decrease in the strength of the boundary current between the mid-1990s and 2015 has been observed in tracer studies"

Please refer to the tracked-changes manuscript for further changes of dates to be more precise.

Line 623 "To this aim" does not make sense; should be e.g. "To this end" or "To achieve

this aim". And especially in the surrounding paragraph (lines 615-624), but also throughout, please check capitalisation (e.g., carbon and argon should not have capital first letters).

Lines 623: changed to "To this end"

Capitalization has been checked and corrected throughout the manuscript.

---

## Author Response (AR3)

**Reviewer 1**

Dear Reviewer 1,

Thank you for your review and helpful suggestions.

Please find the responses to your comments below.

On behalf of all co-authors,

Anne-Marie Wefing

*Note: Reviewer comments in black (lines refer to the original manuscript), response in blue*

General Assessment

This is a well-executed and ambitious study. The methodology is robust and clearly presented, and the figures—both in terms of quality and content—are excellent. However, the manuscript's organization and presentation could be improved. Given the amount of data and analysis presented, these adjustments are non-trivial but will significantly enhance readability. That said, some of these suggestions are inherently subjective.

**Title.** Recommendation: The current title focuses narrowly on Atlantic Waters, but the manuscript covers a broader range of water masses. Consider adjusting the title to reflect this broader scope, even in the abstract, where Atlantic water is only discussed in the final paragraph. Options might include:

Thank you for this comment. We agree that the title was not entirely representative of all the content in the manuscript and thus suggest the following revised title: "Changes in water mass composition and circulation in the central Arctic Ocean between 2011 and 2021 inferred from tracer observations"

**Section 3.**

Terminology: Use terms consistently—e.g., choose either potential temperature or temperature, and salinity or practical salinity throughout.

Changed to "potential temperature" and "practical salinity" everywhere.

Quantification: Reduce the use of vague comparative language such as "around," "slightly more," or "much higher." Instead, refer directly to measured values, as these are available in the figures.

We reduced the use of vague language and refer the reviewer to the tracked-changes document.

Narrative Style: The figures are beautiful and highly informative—consider letting them speak more for themselves. The current text is thorough, but some passages feel overly explanatory, which detracts from readability.

Thank you for your positive feedback on the figures. We carefully considered your suggestion and tried to shorten the description where possible. However, we believe that a certain level of detail remains important to guide readers who may not be familiar with the field of tracer oceanography.

Section 3.3–3.6: These could potentially be moved to the Discussion, as they begin to interpret rather than purely describe.

In the first version of the submitted manuscript, these sections were part of the Discussion. Following the suggestions from reviewer 2 in the first round of reviews, we moved them to Results, so we would like to keep them this way. However, in order to avoid overinterpretation in the Results section, we have now removed some parts that were not purely descriptive .

Idea: It would be helpful to define Sections 1 and 2 by station numbers and geographic location early in the manuscript. This would eliminate the need to repeatedly remind the reader later. Maybe abbreviate to S1 and S2..

Sections 1 and 2 are now defined in Methods (section 2.1) and we shortened/adjusted the text referring to the sections in Results. We decided against an abbreviation, we think it is clearer referring to them as "Section 1" etc.

**Section 4.**

Some subsections (4.1, 4.5, 4.6) feel underdeveloped. Suggest adding a short note on uncertainty due to data resolution. Consider explicitly addressing the uncertainty introduced by the spatial and temporal resolution of the datasets. This would strengthen the interpretations, particularly those related to circulation patterns.

We added some notes on the limited temporal and spatial coverage to sections 4.1 and 4.5. Section 4.6 already includes a note on the better temporal resolution required for addressing long-term changes, which we feel captures the uncertainty due to data resolution ("Also, an extended temporal coverage with high spatial sampling resolution across multiple Arctic basins is probably required for the mid-depth layer, since changes occur more slowly than in the surface layer.").

**Recommendation: Recommended for publication.**

**Specific Suggestions and Comments**

**Lines 1–20**: The abstract would benefit from streamlining and reorganization. Consider introducing both new and historical datasets earlier, as both are essential for analyzing temporal changes — and should only be introduced once. For example, move the sentence starting with "Temporal changes in the circulation [...] between 2011 and 2021" (line 15) closer to the beginning to clarify the study's focus.

The abstract has been changed according to the reviewer's suggestions.

**Line 5**: Remove "mainly" — both datasets are necessary to characterize Atlantic Water (AW) changes.

removed

**Line 28**: Add original citations for the stated information.

We added references to Manabe & Stouffer, 1995, and Rahmstorf, 1996.

**Line 32**: Replace "driven" with "linked to" or "associated with" — warming is likely the true driver.

replaced

**Line 41**: Replace "was" with "is," or clarify the time frame. Consider citing Smith et al. (2021) here, as it discusses AO-related boundary current changes.

replaced

**Line 47**: Use "the two *primary* large-scale circulation patterns."

changed

**Line 49**: Add original references.

We added references to Aagaard & Carmack, 1989, and Carmack, 2000.

**Lines 70–73**: Remove "extensive." Add original sources: e.g., Polyakov et al. (2005) use T anomaly propagation, Dmitrenko et al. (2008), Woodgate et al. (2001), Li et al. (2021). For tracers, include Frank et al. (1998); Broecker & Peng (1982) may not be the best reference for CFCs/$SF_6$.

Removed "extensive". We added references to Woodgate et al. (2001), Polyakov et al. (2005), Dmitrenko et al. (2008), Li et al. (2021) for hydrographic measurements, and to Frank et al. (1998) for gas tracers.

**Figure 1**: Consider color-coding historical data by cruise or sampling year for clarity.

Thank for pointing this out. Figure 1 has been updated, now also including the stations shown in the isosurface map in Figure 6, which were missing before. Different symbols now denote different sampling years.

**Line 89**: Also cite Smith et al. (2011).

Smith et al 2011 did not combine I129 and U236, so we added the reference at the beginning of this paragraph instead, about the general use of anthropogenic radionuclides as Atlantic Water tracers in the Arctic.

**Line 95**: Add a supporting citation.

We added references to Smith et al. (2011) and Wefing et al. (2021).

**Line 98**: Add a reference for the statement about timescales "on the order of decades or below."

We added references to Solomon et al. (2021) and Polyakov et al. (2025).

**Line 103**: Cite a source for the convergence of waters before exiting via Nares or Fram Strait.

We added references to Newton & Sotirin (1997) and de Steur et al. (2013).

**Line 107**: Replace "Arctic system" with a more specific term (e.g., Arctic Ocean circulation or stratification).

Replaced by "circulation in the Arctic"

**Line 139**: "Data from *six* expeditions."

changed

**Lines 142–145**: Use original references.

We added references to Smith et al. (1998, 1999, 2005), Steier et al. (2008), Sakaguchi et al. (2012).

**Lines 158–159**: Remove the phrase referencing Raimondi et al. (2024) and Payne et al. (2024).

removed

**Line 180**: Rephrase to "This model calculates mixing lines…" for clarity.

rephrased

**Line 195**: Clarify whether "mid-depth" here matches the typical depth range used for AW elsewhere in the manuscript.

Yes, this has been clarified.

**Lines 245–249**: Consider revising to: "Station 5 shows a sharp decrease between 700 and 1000 m.

We are not sure to what the reviewer is referring here, since a sharp decrease between 700 and 1000m was not observed for station 5.

**Line 280**: "Color-coded" may be redundant — consider removing.

removed

**Line 306**: "Close to the temperature maximum" — is this referring to the AW core? Clarify.

Yes, this has been clarified.

**Line 319**: Sentence beginning with "The compilation of available 129I data…" may not be necessary — consider cutting.

removed

**Line 330**: Rephrase: "To account for differences in sampling depths and water mass composition…"

rephrased

**Line 333**: "Very low" may be unnecessary — consider deleting.

deleted

**Lines 343–345**: Replace "shallow" with "surface waters" for consistency.

replaced

**Line 365**: [Appears incomplete — please confirm if something's missing.]

We have modified the sentence to "The TTD parameters Γ and Δ, and the derived Δ/Γ ratio and tmode for SAS2021 and JOIS 2020 (Canada Basin) are presented in Fig. 8." and hope this improves readability.

**Line 411**: Clarify which layer is being referred to — "surface" vs. AW could be confusing. Add original Rudels reference.

Sentence changed to: "The circulation times obtained for Atlantic Water in the surface layer in 2021…"

Reference changed to Rudels 2009.

**Line 431**: Rephrase and support with citations: e.g., "Because 129I is introduced via Atlantic waters and diluted by Pacific inflow, its distribution across the SAS2021 section reflects the Atlantic–Pacific water mass composition in PSW."

Changed accordingly and references added.

**Line 531**: Sentence on TTD-derived mode ages not needed in the Discussion — already addressed in Results.

removed

**Lines 564–571**: Quantify terms like "slowdown" and "increasing ages" for accessibility to broader readership.

We added that the slowdown is between 3-8 years, depending on the location and depth.

**Lines 598–600**: Consider omitting this restatement if already covered earlier.

We believe it is still important to repeat this statement here in the Conclusions, even though it was covered in the Results already, since it is an important finding of this study. We shortened the sentence to "Along the transect of the SAS2021 expedition, we observed a sharp drop in radionuclide concentrations in the surface layer between the Amundsen and the Makarov Basins, directly at the Lomonosov Ridge."

**Line 615**: This is the first 129I and 236U dataset north of Greenland — highlight this earlier, perhaps in the abstract.

This has been added to the abstract.

**Line 651**: Add a sentence defining equation terms: e.g., "...where fA, fPac, fSIM, and fMet represent the fractions of Atlantic Water, Pacific Water, sea-ice melt, and meteoric water, respectively."

added

**Reviewer 2**

Dear Reviewer 2,

Thank you for your review and helpful suggestions.

This response letter includes responses to your comments, outlined below.

On behalf of all co-authors,

Anne-Marie Wefing

*Note: Reviewer comments in black (lines refer to the original manuscript), response in blue*

First, I want to apologize for the time I have taken to re-review the manuscript. I should have been faster but my agenda has filled up faster than I could keep up.
Second, I want to thank the authors for the large efforts to restructure the manuscript. I am very honestly impressed and have really not much more to say than that this is excellent work. The little more I would like to be addressed is here:

**Major**

While I get why it is important how the circulation works, wouldn't it be possible to show some implications of changes between Atlantic and Pacific waters? I understand that this shift is not the main part of this paper but it is what you can analyze, in addition to the slowing of the Atlantic water circulation. It would really be helpful to better explain why it is important to know if there is a difference. Does it effect carbon or nutrients, does it affect sea ice, or ocean heat uptake or ecosystems? Basically, if I am not an oceanographer with an interest in the Arctic, it becomes hard to understand why this importance is important beyond the fact that the Arctic is important in general. There might not be space in the abstracts but at the end of the first paragraph of the Introduction, you could explain how changes inside the Arctic Ocean specifically impact the north Atlantic. Which change of which current would cause bad results for Atlantic Overturning, for example. Also, it could be interesting to say what a larger extend of Pacific waters at the surface in the Arctic might mean or what a slowdown of the Atlantic waters at depth could mean.

Thank you for your comment, we have expanded the introduction to include the impacts of changes in the Atlantic-Pacific Water distribution in the surface layer and of changes in circulation in the Atlantic layer. We hope the motivation for this study is now clearer and refer to the tracked-changes version of the manuscript.

**Minor**

Lines 3 and 4: I would suggest reordering the sentence to follow the principle "old before new" by moving the circulation that you study to the beginning of the study and the radionuclides to the end of the sentence. That would then directly link to the next sentence that starts with the radionuclides.

Changed to: "Here, we investigate Atlantic Water circulation in the central Arctic Ocean in 2021 and to assess temporal changes thereof between 2011 and 2021 by using the long-lived anthropogenic radionuclides 129I and 236U."

Line 7 and 8: Similar suggestion, it might be easier to read when you first say what you are looking at and then say with what you are going to look at it.

Changed to: "We obtain tracer ages as well as the mixing of different endmembers in the surface layer from a mixing model."

Line 10: Is significant used in a statistical sense? Otherwise, I'd suggest to replace it by substantial or a similar word.

Changed to "substantial"

Lines 14 to 15: Are we still in waters north of Greenland or is this sentence about halocline waters? I am a bit confused if it means that Pacific waters take more time than Atlantic waters or that Atlantic waters take more time on one side of the ridge than on the other. Please clarify. I think that it would be highly important to say that you speak of Atlantic waters in both cases, and that they take different routes with different transport times.

Changed to: "Circulation times of Atlantic Water in the mid-depth layer point to a longer transport route on the Makarov Basin side of the Lomonosov Ridge compared to the Amundsen Basin."

Lines 16 to 18: Please clarify that you talk about surface waters to avoid confusion.

Changed to: "In the surface layer, we find a shift of the Atlantic-Pacific Water front from the Makarov Basin towards the Lomonosov Ridge from 2011/12 to 2015 and 2021."

Lines 24 to 26: Between physics and ecosystems, there is also the biogeochemistry with changes in acidification and primary production. Maybe worthwhile do add this here.

Added, including a reference to Juranek (2022).

Paragraph 2 of Introduction: Maybe also add a sentence or two on the consquences for the Arctic of this Atlantification in terms of sea ice and biogeochemistry.

We added the following at the end of paragraph 2: With respect to biogeochemistry, the shift to more Atlantic-like conditions might increase nutrient availability and potentially also support projected increases in primary production (Henley et al. 2020).

Paragraphs 2 and 3 of the Intro: Maybe also add something of enhanced freshening and potential consequences.

We added the following at the end of paragraph 2: Further, enhanced freshening of the Arctic and resulting changes in stratification also affect sea-ice growth and primary production (Ardyna and Arrigo, 2020).

Lines 94 and 95: Please indicate a bit more precisely why the surface can be used for radionuclides. I know it is almost obvious but it would not hurt to say that the radionuclides are not mainly entering the ocean from the surface.

Changed to: "In contrast to CFCs and SF6, 129I and 236U can also be used to assess circulation times in the surface layer, which is in contact with the atmosphere, as they are introduced mainly via liquid discharges and not through air-sea gas exchange."

Figure 8: It looks strange to see 3 times AB in the figures on the x-Axis. Maybe add a subscript or something to clarify that each point is another station.

Changed

Line 414: Transport of what? Please try to be more precise to allow for a good understanding

Sentence changed to: Young waters with high tracer concentrations found in the central Amundsen Basin suggest a more direct transport of waters containing the tracer signal compared to the southern Eurasian Basin, potentially as part of the Transpolar Drift

Lines 414-419: If I understand it right, the high age comes from the transport times and the low concentration from both, the transport time and the dilution. However, the dilution does not affect the age as that comes from the mixing line? Wouldn't it be better to separate the sentence to make that clear. I am not even sure if my understanding is correct.

The sentence has been separated into: "The much higher tracer ages (30-50~years) in PSW samples from the Canada Basin (JOIS2020, grey squares in Fig. 7) compared to the SAS2021 samples from the central Arctic are due to the longer travel time of Atlantic Water to the Canada Basin. The higher dilution of the tracer signal in the surface Canada Basin is explained by the fact that mainly Pacific Water is present at the surface."

Further clarification: The ages are obtained from the mixing lines, i.e., the concentration of I129 relative to U236 (changing over time due to different shapes of input functions of both). The dilution is obtained from the absolute concentration of both tracers.

Line 535: You can remove the last sentence and add the references to the sentence before, I think.

removed

---

## Author Response (AR4)

**Response to editor comment on revised version of the manuscript:**

Comment:

Thank you for your updates to the manuscript following the second round of reviewer comments. There is one minor technical issue which is that "psu" has appeared after practical salinity values in the final version, though practical salinity is dimensionless (see e.g. https://tos.org/oceanography/assets/docs/6-3_letter.pdf). Please could you remove the superfluous psus and then I will put the manuscript forward for publication.

Dear Editor,

Thank you very much for pointing this out. We removed "psu" throughout the manuscript and the final files have been uploaded.

On behalf of all co-authors,

Anne-Marie Wefing